# A Single Architecture for Representing Invariance Under Any Space Group

**Cindy Y. Zhang**
Department of Computer Science
Princeton University
cindyz@princeton.edu

**Elif Ertekin**
Department of Mechanical Science & Engineering
University of Illinois Urbana-Champaign
ertekin@illinois.edu

**Peter Orbanz**
Gatsby Computational Neuroscience Unit
University College London
p.orbanz@ucl.ac.uk

**Ryan P. Adams**
Department of Computer Science
Princeton University
rpa@princeton.edu

## Abstract

Incorporating known symmetries in data into machine learning models has consistently improved predictive accuracy, robustness, and generalization. However, achieving exact invariance to specific symmetries typically requires designing bespoke architectures for each group, limiting scalability and preventing knowledge transfer across related symmetries. In the case of the space groups—symmetries critical to modeling crystalline solids in materials science and condensed matter physics—this challenge is particularly salient as there are 230 such groups in three dimensions. In this work we present a new approach to crystallographic symmetries by developing a single machine learning architecture that is capable of adapting its weights automatically to enforce invariance to any input space group. Our approach is based on constructing symmetry-adapted Fourier bases through an explicit characterization of constraints that group operations impose on Fourier coefficients. Encoding these constraints into a neural network layer enables weight sharing across different space groups, allowing the model to leverage structural similarities between groups and overcome data sparsity when limited measurements are available for specific groups. We demonstrate the effectiveness of this approach in achieving competitive performance on material property prediction tasks and performing zero-shot learning to generalize to unseen groups.

## 1 Introduction

Many physical systems are governed by known symmetries, from the arrangement of atoms in a crystal to the laws of motion. By encoding these symmetries—typically described by mathematical groups—directly into a model's architecture, we can enforce that its predictions respect the underlying physical constraints. These group symmetrization techniques act as a powerful inductive bias, leveraging *a priori* knowledge of geometric and physical invariances to enhance model efficiency and reduce training complexity (Thomas et al., 2018; Jing et al., 2020; Satorras et al., 2021; Liu et al., 2023; Du et al., 2023). However, each new type of desired group invariance has generally required a new neural network architecture. In this work we do something different: *we develop a single architecture that can adapt and share structure across the crystallographic groups.*

Crystallographic groups are important in several different domains (Ten Eyck, 1973; Hinuma et al., 2017; Watanabe & Lu, 2018), but most commonly arise in materials science and condensed matter physics, where they describe the transformations that preserve structure in a crystalline solid. The groups are discrete sets of transformations on Euclidean space. They each include translations to capture periodicity, but also include other isometries such as rotations, reflections, glides, inversions, and screws. It is desirable to capture these symmetries in machine learning models for, e.g., building generative models of crystalline solids (Chang et al., 2025; Zeni et al., 2025) and neural ansätze for solving the associated electronic Schrödinger equation (Li et al., 2022; Huang et al., 2025).

A challenge of symmetrized machine learning models for crystallographic groups, however, is data sparsity: there are 230 groups in three dimensions and even the most commonly-used materials benchmark (Jain et al., 2013) has only about 200,000 data points, averaging *fewer than 1,000 examples per group*. In practice, the data distribution is heavily skewed, with many groups having few or no examples for certain properties. Thus developing a separate machine learning architecture for each group is unlikely to be successful; rather we must find a way to inform our models with group symmetry while also sharing parameters across the different groups. We develop such an architecture in this work and show that it can generalize even to groups it has not been trained on.

Our approach centers on a new algorithm for constructing symmetry-adapted Fourier bases for the crystallographic groups. We analytically derive the explicit constraints imposed by crystallographic symmetries on the Fourier coefficients of functions defined over Euclidean space. We prove that this symmetry-adapted Fourier basis spans the space of continuous, square-integrable functions invariant to the given crystallographic group. This allows us to represent crystal structures linearly using atom positions expanded in the symmetry-adapted Fourier basis.

Crystalline materials are typically represented as graphs within ML pipelines, which need to be augmented with additional edges or features to encode the periodic lattice structure explicitly (Xie & Grossman, 2018; Yan et al., 2022). In our framework, rather than separately augmenting each graph with information about the crystal lattice, we have a layer that effectively encodes positions in the standard Fourier basis and then multiplies the encodings by a precomputed, group-dependent adjacency matrix that defines the constraints between Fourier coefficients. This symmetry-adapted encoding can serve as input to further network layers to produce a crystallographic group-invariant output, allowing the parameters of the neural network to be shared across all space groups.

We empirically verify the effectiveness of this approach in learning positional encodings within crystal structures that accurately reflect orbit distances. We further validate our framework by using these positional encodings in a Transformer model, improving performance over standard positional encodings and achieving competitive performance across several benchmark tasks for material property prediction. We also show empirically that the approach can perform zero-shot learning and generalize across groups for which it has never seen data. Our framework can be flexibly used as an encoding module that integrates with existing ML models to capture the exact symmetries of crystallographic data.

The paper is structured as follows. We start by defining crystallographic groups and presenting the intuition from one-dimensional Fourier series that extends to a Fourier basis for functions invariant to crystallographic groups. We then derive the exact constraints on Fourier coefficients in Section 3.1 and their dual graph representation in Section 3.2. The adjacency matrices of these graphs are key to the architecture introduced in Section 3.3 that adapts its invariance properties to different crystallographic groups. We contextualize our findings with a discussion of related work in Section 4. The details of our experiments and results are in Section 5.

## 2 BACKGROUND

In this section, we describe invariance under crystallographic groups and define the $G$-invariant functions of interest. We then present a key result from Adams & Orbanz (2023) proving that a Fourier representation exists for continuous $G$-invariant functions.

### 2.1 CRYSTALLOGRAPHIC GROUP INVARIANCE

Crystallographic groups are discrete groups of isometries that tile Euclidean space with a repeating fundamental region. We consider the fundamental region $\Pi$ to be a convex polytope. Copies of $\Pi$ under the group's isometries fill $\mathbb{R}^n$ without gaps or overlaps. Only a finite number of crystallographic groups exist in each dimension; there are 17 such groups on $\mathbb{R}^2$, called the *wallpaper groups*, and 230 such groups on $\mathbb{R}^3$, called the *space groups*, which classify all possible symmetries of crystals. The tiling patterns of the wallpaper groups are shown in Appendix A.1.

Each isometry $\phi$ in a crystallographic group $G$ takes the form $\phi(\boldsymbol{x}) = \mathbf{A}\boldsymbol{x} + \boldsymbol{t}$, where $\mathbf{A} \in \mathbb{R}^{n \times n}$ is an orthonormal matrix and $\boldsymbol{t} \in \mathbb{R}^n$ is a translation vector. By definition, $G$ contains translations by any integer combination of $n$ linearly independent basis vectors $\boldsymbol{\tau}_1, \ldots, \boldsymbol{\tau}_n$, making it both discrete

and infinite. A function $f : \mathbb{R}^n \to \mathbb{R}$ is $G$-invariant if it satisfies:

$$f(\phi(\boldsymbol{x})) = f(\boldsymbol{x}) \qquad \text{for all } \phi \in G \text{ and } \boldsymbol{x} \in \mathbb{R}^n. \tag{1}$$

$G$-invariance is crucial in modeling physical phenomena in crystals. According to Neumann's principle, if a physical system is invariant with respect to the symmetries of a group $G$, then any physical observables of that system must also be invariant to the same symmetries (Bradley & Cracknell, 2009). Formally, this invariance implies that $f$ can be expressed solely in terms of equivalence classes of points, known as orbits, defined by the group action. The *orbit* of a point $\boldsymbol{x} \in \mathbb{R}^n$ is the set $\{\phi(\boldsymbol{x}) \mid \phi \in G\}$ of all points mapping to it under the group's symmetries. The quotient set $\mathbb{R}^n / G$ has a continuous bijective mapping to the fundamental region $\Pi$, which contains exactly one point from each orbit. Because an invariant function is fully determined by its values on the fundamental region $\Pi$, this region serves as a natural domain for analysis.

## 2.2 FOURIER REPRESENTATIONS

In order to construct $G$-invariant functions, we want a set of basis functions that are periodic. One natural choice is a Fourier series. In the one-dimensional setting, a smooth function $f : \mathbb{R} \to \mathbb{R}$ of period $L$ admits the well–known expansion

$$f(x) = a_0 + \sum_{k=1}^{\infty} a_k \cos(2\pi k x / L) + b_k \sin(2\pi k x / L).$$

These sine and cosine modes are exactly the eigenfunctions of the one–dimensional Laplacian,

$$-\frac{d^2}{dx^2} e = \lambda e \quad \text{subject to} \quad e(x + L) = e(x), e'(x + L) = e'(x),$$

with eigenvalues $\lambda = (2\pi k / L)^2$. Two key insights carry forward to crystallographic groups: (1) Translation invariance imposes boundary conditions that yield a discrete spectrum of Laplace eigenfunctions, and (2) the resulting eigenfunctions form an orthonormal basis for all square-integrable $L$-periodic functions.

Adams & Orbanz (2023) show that for any crystallographic group $G$, there exists a sequence of $G$-invariant functions $e_1, e_2, \ldots$ on $\mathbb{R}^n$ such that any $G$-invariant function in $L_2(\Pi)$ can be represented as a linear combination of these basis functions. This basis generalizes the standard Fourier series to account for all symmetries in a crystallographic group, not just translations. The work characterizes these basis functions, $e_i$, as the solutions to the constrained partial differential equation

$$-\Delta e = \lambda e$$
$$\text{subject to} \qquad e = e \circ \phi \text{ for all } \phi \in G. \tag{2}$$

While this result establishes the existence of a $G$-invariant Fourier basis, the construction in Adams & Orbanz (2023) relies on numerical methods to approximate the eigenfunctions of the Laplace operator. We introduce an alternative, analytical approach in the following section.

## 3 METHODS

In this section, we present the theoretical and algorithmic foundations for our adaptive framework. First, we derive the analytical constraints that invariance to a crystallographic group imposes on a function's Fourier series coefficients. Second, we show how these constraints define a complete, $G$-invariant basis, which admits a dual representation in terms of graphs. These graphs allow us to formalize a constructive algorithm for the $G$-invariant basis. Finally, we use these basis functions to define the *Crystal Fourier Transformer (CFT)*, a space group adaptive architecture.

## 3.1 DERIVING GROUP CONSTRAINTS IN FOURIER SPACE

Our goal is to find a universal representation for the constraints imposed by invariance to a crystallographic group $G$. We achieve this by analyzing the effect of group operations on the Fourier transform of a $G$-invariant function $f : \mathbb{R}^n \to \mathbb{R}$.

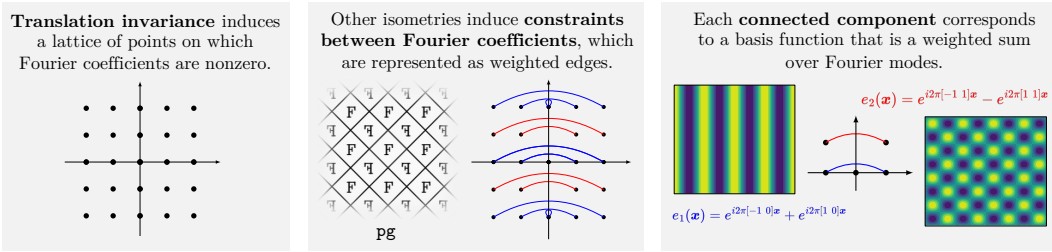

Figure 1: We construct the Fourier basis for $G$-invariant functions using a dual representation in terms of graphs. The translations in a crystallographic group $G$ give rise to a discrete lattice of points corresponding to the Fourier series. We represent constraints induced by additional isometries as weighted edges between these points. For wallpaper group pg, which contains the translations and glide reflections present in the tiling pattern, we get edges of weight $-1$ (red) and $+1$ (blue). Each clique in the resulting graph can be expanded as one of the pictured Fourier basis functions.

The translation symmetries present in any crystal structure require that $f$ be periodic. This periodicity constrains the support of its Fourier transform, $F(\boldsymbol{\omega})$, to a discrete set of frequencies known as the *reciprocal lattice*, $\mathcal{L}^*$. For any frequency $\boldsymbol{\omega}$ not on this lattice $F(\boldsymbol{\omega})$ must be zero.

The non-translation isometries in $G$ (e.g., rotations, reflections, glides) impose additional constraints that relate the values of the Fourier coefficients at different points on the reciprocal lattice. The following proposition makes this relationship precise.

**Proposition 3.1.** *Let $f : \mathbb{R}^n \to \mathbb{R}$ be a $G$-invariant function and let $F(\boldsymbol{\omega})$ be its Fourier transform. All symmetry operations $\phi \in G$ have the form $\phi(\boldsymbol{x}) = \mathbf{A}\boldsymbol{x} + \boldsymbol{t}$, where $\mathbf{A}$ is an orthogonal matrix and $\boldsymbol{t}$ is a translation vector. For any $\phi \in G$, the Fourier coefficients must satisfy the following relation for all $\boldsymbol{\omega} \in \mathcal{L}^*$:*

$$F(\boldsymbol{\omega}) = e^{i2\pi\boldsymbol{\omega}^\top \mathbf{A}^\top \boldsymbol{t}} F(\mathbf{A}\boldsymbol{\omega}). \tag{3}$$

The proof is in Appendix A.2. Equation 3 states that the Fourier coefficients at frequencies $\boldsymbol{\omega}$ and $A\boldsymbol{\omega}$ are coupled by a phase factor $e^{i2\pi\boldsymbol{\omega}^\top \mathbf{A}^\top \boldsymbol{t}}$ determined by the specific isometry $\phi$. These constraints partition the reciprocal lattice into orbits of related frequencies and define the exact structure of the invariant basis functions, as formalized in our main result.

**Theorem 3.2.** *Let $G$ be a crystallographic group and $\mathcal{L}^*$ be its reciprocal lattice. The linear components $(\mathbf{A}_\phi)$ of the group's transformations $(\phi(\boldsymbol{x}) = \mathbf{A}_\phi\boldsymbol{x} + \boldsymbol{t}_\phi)$ partition $\mathcal{L}^*$ into disjoint orbits $\mathcal{O}$, where frequencies $\boldsymbol{\omega}_1, \boldsymbol{\omega}_2 \in \mathcal{O}$ if $\boldsymbol{\omega}_2 = \mathbf{A}_\phi\boldsymbol{\omega}_1$ for some $\phi \in G$.*

*Assume an orbit $\mathcal{O}$ is phase-consistent, meaning for any $\boldsymbol{\omega} \in \mathcal{O}$ and any transformation $\phi$ that maps $\boldsymbol{\omega}$ to itself (i.e., $\mathbf{A}_\phi\boldsymbol{\omega} = \boldsymbol{\omega}$), the corresponding phase factor $e^{i2\pi\boldsymbol{\omega}^\top \mathbf{A}_\phi^\top \boldsymbol{t}_\phi}$ is equal to 1.*

*Then, a corresponding basis function $e_\mathcal{O}(\boldsymbol{x})$ can be constructed as:*

$$e_\mathcal{O}(\boldsymbol{x}) = \sum_{\boldsymbol{\omega}\in\mathcal{O}} w_{\boldsymbol{\xi}\to\boldsymbol{\omega}} \cdot e^{i2\pi\boldsymbol{\omega}^\top \boldsymbol{x}} \tag{4}$$

*where the complex coefficients $w_{\boldsymbol{\xi}\to\boldsymbol{\omega}}$ are uniquely determined by the phase constraint in equation 3, relative to an arbitrary reference frequency $\boldsymbol{\xi} \in \mathcal{O}$ for which $w_{\boldsymbol{\xi}\to\boldsymbol{\xi}} := 1$. Every continuous, $G$-invariant function $f$ admits a uniformly convergent expansion*

$$f = \sum_\mathcal{O} c_\mathcal{O} e_\mathcal{O},$$

*where the family $\{e_\mathcal{O}\}$ is constructed over all phase-consistent orbits in $\mathcal{L}^*$.*

The proof is in Appendix A.3. Theorem 3.2 transforms the problem of constructing $G$-invariant functions from a continuous problem on $\mathbb{R}^n$ to a discrete problem on the reciprocal lattice. Each basis function is a specific linear combination of Fourier modes, where the coefficients are uniquely determined by the group's symmetry constraints. The phase-consistency condition filters out frequencies that cannot support a globally consistent invariant function.

While Theorem 3.2 provides the analytical form of the basis, it does not specify a computational procedure for identifying these orbits and their coefficients. To operationalize this, we introduce a dual graph representation that makes the group constraints concrete and computationally tractable.

---

**Algorithm 1** Symmetry-Adapted Fourier Basis Construction

---

**Require:** Space group $G$, reciprocal lattice $\mathcal{L}^*$
1: Construct constraint graph $\mathcal{G}$ with nodes $\mathcal{L}^*$ and edges $\boldsymbol{\omega} \to \mathbf{A}\boldsymbol{\omega}$ weighted by $e^{i2\pi\boldsymbol{\omega}^\top \mathbf{A}^\top \boldsymbol{t}}$ for each $\phi(\boldsymbol{x}) = \mathbf{A}\boldsymbol{x} + \boldsymbol{t} \in G$
2: Remove nodes with inconsistent self-loops (weight $\neq 1$) from $\mathcal{G}$
3: Identify phase-consistent orbits $\{\mathcal{O}_k\}$ as connected components of $\mathcal{G}$
4: Construct basis functions $e_{\mathcal{O}_k}(\boldsymbol{x}) = \sum_{\boldsymbol{\omega} \in \mathcal{O}_k} w_{\boldsymbol{\xi} \to \boldsymbol{\omega}} \cdot e^{i2\pi\boldsymbol{\omega}^\top \boldsymbol{x}}$ where $w_{\boldsymbol{\xi} \to \boldsymbol{\omega}}$ is the product of edge weights along any path from node $\boldsymbol{\xi}$ to node $\boldsymbol{\omega}$
5: **return** Basis $\{e_{\mathcal{O}_k}\}$

---

## 3.2 Dual Graphs of Crystallographic Fourier Series

The algebraic relationships and phase constraints from Proposition 3.1 can be captured as a directed graph on the reciprocal lattice. In this graph, nodes represent frequencies $\boldsymbol{\omega} \in \mathcal{L}^*$, and each symmetry operation $\phi(\boldsymbol{x}) = \mathbf{A}\boldsymbol{x} + \boldsymbol{t}$ induces a set of directed edges. For each $\boldsymbol{\omega}$, an edge is drawn from $\boldsymbol{\omega}$ to $\mathbf{A}\boldsymbol{\omega}$ with a complex weight equal to the phase factor $e^{i2\pi\boldsymbol{\omega}^\top \mathbf{A}^\top \boldsymbol{t}}$.

This graph structure is a key computational tool for encoding the basis functions. The phase-consistent orbits of Theorem 3.2 correspond exactly to the connected components of this graph after nodes with inconsistent self-loops are removed. The basis function coefficients $w_{\boldsymbol{\xi} \to \boldsymbol{\omega}}$ are the product of edge weights along any path from a reference node $\boldsymbol{\xi}$ to another node $\boldsymbol{\omega}$ within a component. This framework leads directly to Algorithm 1 for constructing the full symmetry-adapted basis for any space group $G$. Since the reciprocal lattice $\mathcal{L}^*$ is infinite, we in practice construct the reciprocal lattice up to a maximum frequency radius, yielding a finite set of basis functions that form a practical approximation to the complete basis.

**Example of basis construction.** We describe the construction of the constraint graph and the corresponding basis for the wallpaper group pg, which consists of translations and glide reflections. For simplicity, we'll use the standard basis vectors $\boldsymbol{e}_1 = \begin{bmatrix} 1 & 0 \end{bmatrix}^\top$ and $\boldsymbol{e}_2 = \begin{bmatrix} 0 & 1 \end{bmatrix}^\top$ for the translations. Invariance to these standard basis shifts induces the reciprocal lattice $\mathbb{Z}^2$ where Fourier coefficients are nonzero. Next we incorporate glide reflections, given by the form

$$\phi(\boldsymbol{x}) = \begin{bmatrix} -1 & 0 \\ 0 & 1 \end{bmatrix} \boldsymbol{x} + \begin{bmatrix} 0 \\ 1/2 \end{bmatrix}, \tag{5}$$

which are a combination of a reflection $\mathbf{A}$ and a half-shift $\boldsymbol{t}$. We calculate the weight for the edge from a frequency $\boldsymbol{\omega}$ to its reflected counterpart $\mathbf{A}\boldsymbol{\omega}$ using the phase formula from Proposition 3.1:

$$\exp\left( i2\pi \begin{bmatrix} \omega_1 & \omega_2 \end{bmatrix} \begin{bmatrix} -1 & 0 \\ 0 & 1 \end{bmatrix} \begin{bmatrix} 0 \\ 1/2 \end{bmatrix} \right) = \exp(i\pi\omega_2). \tag{6}$$

This result provides a simple rule for the edge weights: the weight is $+1$ if $\omega_2$ is even and $-1$ if $\omega_2$ is odd. This pattern of alternating positive (blue) and negative (red) edge weights is exactly what is visualized in Figure 1.

Each $\boldsymbol{\omega}$ corresponds to the Fourier mode $e^{i2\pi\boldsymbol{\omega}^\top \boldsymbol{x}}$. We form the basis functions by taking a weighted sum of the Fourier modes in each connected component. For example, the component containing $(-1, 0)$ and $(1, 0)$ has an edge of weight $+1$ in both directions between the two points. This corresponds to the basis function

$$e_1(\boldsymbol{x}) = e^{i2\pi[-1\ 0]\boldsymbol{x}} + e^{i2\pi[1\ 0]\boldsymbol{x}}.$$

The component containing $(-1, 1)$ and $(1, 1)$ has an edge of $-1$ in both directions. This corresponds to the basis function

$$e_2(\boldsymbol{x}) = e^{i2\pi[-1\ 1]\boldsymbol{x}} - e^{i2\pi[1\ 1]\boldsymbol{x}}.$$

This $G$-invariant basis construction continues for each connected component in the constraint graph, up until some finite radius cutoff. The information needed to write out the analytic form of the basis functions lies in the edge weights, i.e., the adjacency matrix, of the graph.

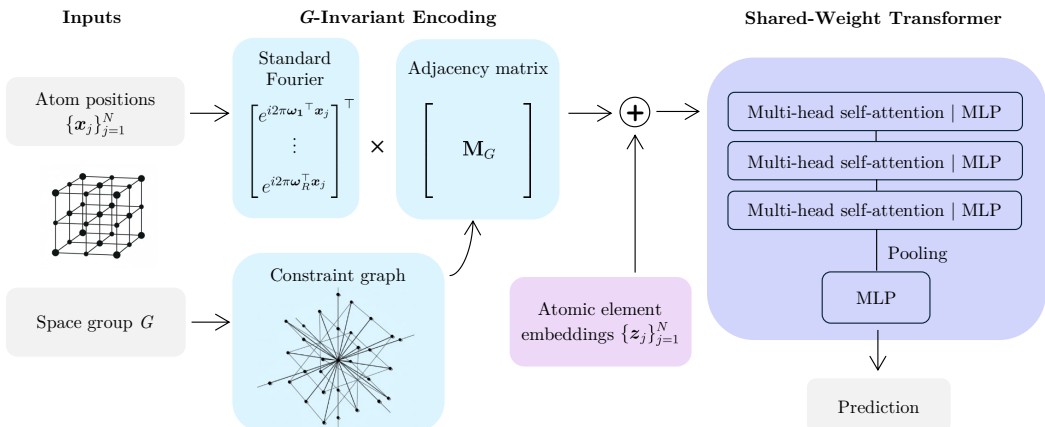

Figure 2: Diagram of the Crystal Fourier Transformer architecture. Atom positions are first encoded into standard Fourier modes. A group-conditional routing matrix, $\mathbf{M}_G$ transforms these modes into a provably invariant basis. These adaptive positional encodings, combined with other invariant features, are then processed by a Transformer whose weights are shared across all space groups to predict material properties.

### 3.3 CRYSTAL FOURIER TRANSFORMER

The construction of the constraint graph for the symmetry-adapted Fourier basis provides the backbone for a novel neural architecture that learns group-invariant representations adaptively. We introduce the *Crystal Fourier Transformer (CFT)*, a model designed to process crystallographic data by explicitly building in invariance to its space group symmetry. The core of CFT is an encoding module that transforms atomic coordinates into a feature representation that is invariant to the specified group $G$.

This transformation is a two-step process, visualized in Figure 2. First, for an input coordinate $\boldsymbol{x}$ and a set of reciprocal lattice frequencies $\{\boldsymbol{\omega}_k\}_{k=1}^R$, we compute a vector of standard Fourier modes, $\mathbf{v}(\boldsymbol{x}) \in \mathbb{C}^R$, where $[\mathbf{v}(\boldsymbol{x})]_k = e^{i2\pi\boldsymbol{\omega}_k^\top \boldsymbol{x}}$. This vector itself is not invariant. Second, we apply our key architectural innovation: a linear transformation defined by the pre-computed adjacency matrix $\mathbf{M}_G$ of the dual graph for the specified group $G$. The result is a vector of symmetry-adapted basis functions evaluated at $\boldsymbol{x}$:

$$\mathbf{e}_G(\boldsymbol{x}) = \mathbf{M}_G \mathbf{v}(\boldsymbol{x}). \tag{7}$$

The matrix $\mathbf{M}_G$ acts as a *group-conditional routing mechanism*. It linearly combines the standard Fourier modes according to the exact constraints dictated by the group $G$, effectively projecting the input representation onto the $G$-invariant basis. This mechanism allows a single, fixed downstream architecture to achieve invariance to any of the 230 space groups simply by swapping the corresponding routing matrix $\mathbf{M}_G$.

CFT leverages this encoding module within a standard encoder-only Transformer architecture (Vaswani et al., 2017), where the self-attention mechanism allows for the modeling of complex interactions between atoms in a crystal. For each atom in a unit cell, we pass the atom position and the space group of the crystal into the encoding module, the only group-dependent component of the model, and we produce an output embedding that is $G$-invariant and contains symmetry-aware positional information.

The $G$-invariant positional encoding is directly added to a standard embedding of the atom's chemical element to form the input token for each atom. The resulting sequence of tokens is processed by the Transformer encoder. Because the input features are already $G$-invariant, any function applied to them, including the self-attention mechanism, will produce a $G$-invariant output. This design allows the core parameters of the Transformer to be *shared across all 230 space groups*, enabling the model to generalize effectively even for groups with sparse data. More generally, the output of the $G$-invariant encoding layer can be used as input to any ML model, and the model will become invariant to the input space group while keeping its own weights shared across all space groups.

## 4 RELATED WORK

**Group Equivariance and Invariance in ML.** Symmetries have a long history in machine learning, dating back to work by Minsky & Papert (1988) discussing invariance in single-layer perceptrons when summing over a finite group of symmetries. This was then extended to multi-layer perceptrons by Shawe-Taylor (1994), who showed that invariant MLPs can be constructed by partitioning the connections between layers into weight-sharing orbits. The introduction of translation invariance in Convolutional Neural Networks (CNNs) (LeCun et al., 1989) was a breakthrough in computer vision, capturing the symmetries of image data. Many works (Cohen & Welling, 2016a; Kondor et al., 2018; Cohen et al., 2019; Finzi et al., 2020; Zhdanov et al., 2024) have generalized the CNN architecture to be equivariant under additional group symmetries, and it was proven in Kondor & Trivedi (2018) that convolutional structure is necessary for equivariance to actions of compact groups.

**Symmetries in AI for Science.** For tasks dealing with the symmetries of the physical world, incorporating equivariance to SE(3) and E(3) symmetries has consistently improved performance and data efficiency (Cohen & Welling, 2016b; Cohen et al., 2018; Fuchs et al., 2020; Geiger & Smidt, 2022; Du et al., 2023). These architectures have found applications across geometry, physics, and chemistry, including protein structure classification (Weiler et al., 2018), learning interatomic potentials and force fields (Batatia et al., 2022; Batzner et al., 2022), 3D point cloud segmentation (Guo et al., 2020), and molecular conformer generation (Jing et al., 2020). These architectures are tailored to specific compact symmetry groups, leveraging spherical harmonics and tensor products of the irreducible representations (Worrall et al., 2017; Thomas et al., 2018). While these models capture the symmetries of 3D space, they are unable to capture the infinite discrete symmetries inherent to crystallographic structures.

**Learning Crystal Representations.** Crystalline materials are typically represented as graphs within ML pipelines. Since initial work on crystal graph representations by Xie & Grossman (2018) and Chen et al. (2019), which use atoms as nodes and bonds as edges, many follow-up works have explored different methods of incorporating additional spatial information and symmetries in the graph representations (Choudhary & DeCost, 2021; Yan et al., 2022; Ruff et al., 2024; Wang et al., 2024; Yan et al., 2024a;b). A separate line of work focuses on creating canonical, unambiguous crystal representations to ensure that models learn from consistent data inputs (Widdowson & Kurlin, 2022; Nigam et al., 2024). These methods differ from our work in a key aspect. While they implicitly handle or are designed for specific symmetries, the Crystal Fourier Transformer is a single, adaptive architecture that enforces exact $G$-invariance for any of the 230 space groups.

## 5 EXPERIMENTS

Our experiments investigate three central questions: (1) Does our symmetry-adapted encoding module learn geometrically meaningful representations that accurately capture the underlying orbit space of a crystal? (2) How does CFT perform on standard, large-scale material property prediction benchmarks against state-of-the-art graph-based models? (3) Does the adaptive nature of our architecture enable effective zero-shot generalization to space groups unseen during training? We demonstrate that CFT learns correct, symmetry-aware distance metrics, achieves competitive performance on benchmark tasks, and generalizes robustly. Full implementation details for the experiments can be found in Appendix A.5.

### 5.1 EXPERIMENTAL SETUP

**Dataset and Task.** For our primary benchmark experiments, we use data from the Materials Project, one of the largest databases of computational material properties (Jain et al., 2013). The task is to predict four key material properties from the crystal structure: total energy (eV/atom), band gap (eV), bulk modulus (log GPa), and shear modulus (log GPa). For each material, the model receives the atomic numbers and fractional coordinates of atoms within the unit cell, the $3 \times 3$ lattice vectors, and the corresponding space group identifier (1-230). At the time of training, there were 152,149 crystals with known total energies and band gaps, and 11,997 crystals with known bulk and shear moduli in the Materials Project database.

**Baselines.** We compare CFT against state-of-the-art graph neural network (GNN) architectures designed for crystal structures. These models represent a fundamentally different approach, relying

on graph representations of the unit cell and its neighborhood instead of our Fourier basis encoding. Our baselines include CGCNN (Xie & Grossman, 2018), ALIGNN (Choudhary & DeCost, 2021), and Matformer (Yan et al., 2022). We also compare against the same Transformer architecture but with the standard sine and cosine positional encodings from (Vaswani et al., 2017) for each dimension, rather than the proposed $G$-invariant encoding, to isolate the effect of our proposed encoding module. All baseline models were re-trained and evaluated under identical conditions for fair comparison. The numbers may differ from the results reported in the original papers due to the Materials Project dataset having greatly increased in size in recent years.

## 5.2 SYMMETRY-ADAPTED ENCODINGS CAPTURE ORBIT DISTANCE

A fundamental assumption of our work is that the $G$-invariant encoding module can learn representations that capture the true geometric structure of the orbit space. We first validate this capability in a controlled, self-supervised setting before applying the model to downstream tasks. The goal is to train the encoder to produce positional encodings where the Euclidean distance between embeddings directly corresponds to the true orbit distance $d_G(\boldsymbol{x}_1, \boldsymbol{x}_2)$ between atoms. The orbit distance is defined as

$$d_G(\boldsymbol{x}_1, \boldsymbol{x}_2) := \min_{\phi_1, \phi_2 \in G} ||\phi_1(\boldsymbol{x}_1) - \phi_2(\boldsymbol{x}_2)||_2. \tag{8}$$

It measures the shortest Euclidean distance between any two points in the respective orbits of the atoms, which stays invariant to applications of group actions to either atom.

**Setup.** We construct a synthetic dataset containing 100,000 samples for each of the 230 space groups. Each sample consists of a pair of random atomic positions as fractional coordinates in $[0, 1)^3$, a space group, and randomly generated lattice vectors that satisfy the constraints of the Bravais lattice for the group. The model takes as input the atomic position and lattice vectors in parallel branches. The position is passed through the $G$-invariant encoding module and a subsequent ResNet (He et al., 2016) to encode symmetry information, while the concatenated lattice vectors are passed through a ResNet to encode scale information. The element-wise product of the outputs of the two ResNets is the final positional encoding. The model is trained to minimize the Mean Squared Error (MSE) between the L2 distance of the output embeddings and the ground-truth orbit distance.

**Results.** The trained model achieves a test Mean Absolute Error (MAE) of 0.102 Å against an average orbit distance of 2.724 Å in the synthetic dataset. This low error means that the embeddings accurately capture the complex, non-Euclidean orbit distance metric across all 230 space groups. We include a qualitative case study of the embeddings for the 2D wallpaper group p6m in Appendix A.6, for which the embedding map is provably correct.

## 5.3 MATERIAL PROPERTY PREDICTION

Having validated our encoding module, we now evaluate the full CFT architecture on the task of predicting material properties from the Materials Project dataset. This experiment assesses CFT's ability to compete with specialized GNNs that have been the dominant paradigm for this task.

**Setup.** We use the pre-trained positional encoding model from Section 5.2 to initialize the positional encodings in CFT. This initialization captures the physical scale of the crystal in this feature space, which is important because the Fourier-based encoding is dimensionless. The positional encoding is directly added to a learned embedding for the atom's chemical element, which forms the input tokens to a standard Transformer encoder with three multi-head self-attention layers with eight heads each, followed by mean pooling and a MLP that outputs the final prediction. Because all inputs are $G$-invariant, the Transformer's weights are shared across all 230 space groups, allowing the model to leverage data from common groups to improve predictions for rare ones.

**Results.** Table 1 shows the test MAE for CFT and baseline models. Our CFT model achieves competitive performance across all four properties and outperforms all baselines on Total Energy and Shear Moduli prediction. Notably, it significantly surpasses the Transformer baseline that uses standard positional encodings, demonstrating the usefulness of the symmetry-adapted encoding. CFT outperforms CGCNN across all properties and is on par with ALIGNN and Matformer.

CFT uses a fundamentally different approach from the graph neural networks, which explicitly encode distances and periodic boundary conditions in the edges (Yan et al., 2022) and encode additional geometric features such as bond angles (Choudhary & DeCost, 2021). CFT relies on the

| Method | Total Energy eV/atom ↓ | Band Gap eV ↓ | Bulk Moduli log GPa ↓ | Shear Moduli log GPa ↓ |
|---|---|---|---|---|
| CFT | $\mathbf{0.197 \pm 0.009}$ | $0.306 \pm 0.006$ | $\mathbf{0.082 \pm 0.008}$ | $\mathbf{0.158 \pm 0.011}$ |
| Transformer | $0.220 \pm 0.019$ | $0.440 \pm 0.014$ | $0.142 \pm 0.012$ | $0.226 \pm 0.016$ |
| CGCNN | $0.453 \pm 0.014$ | $0.343 \pm 0.074$ | $0.093 \pm 0.012$ | $0.178 \pm 0.007$ |
| ALIGNN | $\mathbf{0.203 \pm 0.005}$ | $0.235 \pm 0.002$ | $\mathbf{0.076 \pm 0.002}$ | $\mathbf{0.160 \pm 0.009}$ |
| Matformer | $0.371 \pm 0.011$ | $\mathbf{0.213 \pm 0.003}$ | $\mathbf{0.074 \pm 0.002}$ | $0.179 \pm 0.003$ |

Table 1: Test Mean Absolute Error (MAE) comparisons on the Materials Project dataset. Lower values indicate better performance. Results are reported as mean $\pm$ one standard deviation over 4 runs. **Bold** indicates the best performance for each property.

positional encoding module to capture meaningful geometric information and on the multi-head attention layers to capture complex interactions between atoms.

**Efficiency.** A key advantage of our framework is its computational efficiency. Graph-based models typically require constructing neighborhood graphs and performing iterative message passing, which can be computationally intensive and difficult to parallelize. CFT's core symmetry operation is a single matrix-vector product with the precomputed routing matrix $\mathbf{M}_G$. This, combined with the inherent parallelizability of the Transformer architecture, leads to significant speedups. This performance advantage holds even with a larger parameter budget. The empirical results in Table 2 show CFT to be substantially faster than ALIGNN and Matformer in both training and inference.

| Method | Parameters | Training time (per epoch) | Inference time (total) |
|---|---|---|---|
| CFT | 5.34M | 91 sec | 60 sec |
| ALIGNN | 4.03M | 592 sec | 451 sec |
| Matformer | 2.78M | 266 sec | 222 sec |

Table 2: Training and inference time comparison. Training time is measured per epoch on a dataset of 120k crystals. Total inference time is measured on a dataset of 10k crystals. Amortized pre-processing costs, including the pretraining of CFT's positional encoding module and the one-time graph construction for ALIGNN and Matformer, are not included. All models were benchmarked on a single NVIDIA L40 GPU.

## 5.4 ZERO-SHOT GENERALIZATION TO UNSEEN SPACE GROUPS

The core hypothesis behind our adaptive architecture is that, by explicitly parameterizing the group constraints, CFT can generalize to materials from previously unseen space groups. We test this hypothesis in a zero-shot setting by holding out all space groups containing inversion symmetry from the training set.

**Setup.** Let $\mathcal{G}_{\text{inv}}$ denote the set of space groups with inversion symmetry (centrosymmetric groups). This subset accounts for approximately $49\%$ of the bulk/shear modulus data and $25\%$ of the total-energy data. For each architecture we train two models: (1) *Zero-shot:* trained only on the 138 non-centrosymmetric space groups $\mathcal{G} \setminus \mathcal{G}_{\text{inv}}$, and (2) *All-data:* trained on the full dataset of 230 space groups. Both models are then evaluated on the same dataset of held-out inversion groups. For CFT in the zero-shot regime, we simply provide at inference time the precomputed routing matrix $\mathbf{M}_G$ corresponding to the unseen group $G$; no parameter updates or fine-tuning are performed.

**Results.** For each held-out inversion group $G \in \mathcal{G}_{\text{inv}}$, we compute the group-wise MAE under the zero-shot and all-data training regimes, denoted $\text{MAE}_G^{\text{zero-shot}}$ and $\text{MAE}_G^{\text{all-data}}$. We define the *performance gap* for group $G$ as

$$\Delta\text{MAE}_G = \text{MAE}_G^{\text{zero-shot}} - \text{MAE}_G^{\text{all-data}}. \qquad (9)$$

Figure 3 plots these signed gaps $\Delta\text{MAE}_G$ for all held-out groups with at least 20 data points for both shear modulus and total energy; values close to zero indicate that the zero-shot model matches the fully supervised model on that group. We summarize these results with the *group-balanced absolute performance gap*

$$\text{GB-Gap} = \frac{1}{|\mathcal{G}_{\text{inv}}|} \sum_{G \in \mathcal{G}_{\text{inv}}} |\Delta\text{MAE}_G|. \qquad (10)$$

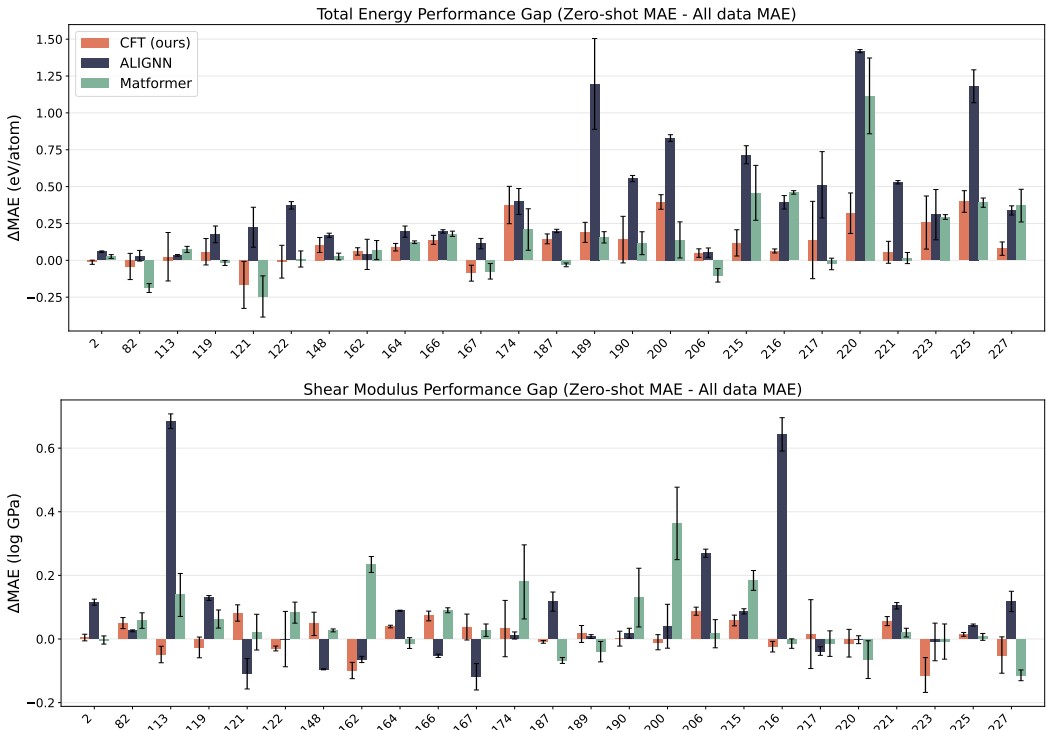

Figure 3: We evaluate the zero-shot generalization capability of CFT in predicting the shear modulus and total energy of materials from groups containing inversion symmetry. For each held-out group, we plot the performance gap $\Delta\mathrm{MAE}$ (Eq. 9), the difference between the zero-shot MAE (trained without any data from inversion groups) and the all-data MAE (trained on all 230 space groups). Smaller values are better; $\Delta\mathrm{MAE} = 0$ corresponds to perfect zero-shot generalization to that group.

This metric first computes an MAE per space group and then averages across groups, so each held-out group contributes equally regardless of its size. For shear modulus, the GB-Gap is $0.042$ log GPa for CFT, compared to $0.120$ for ALIGNN and $0.080$ for Matformer. For total energy, the GB-Gap is $0.141$ eV/atom for CFT, versus $0.309$ for ALIGNN and $0.197$ for Matformer. CFT thus exhibits the smallest average per-group degradation when moving from the all-data regime to zero-shot generalization.

As shown in Figure 3, CFT maintains small gaps across all held-out groups, whereas ALIGNN and Matformer exhibit pronounced failure modes on several groups. In particular, for groups 113, 162, 200, and 216 the zero-shot shear-modulus MAE of ALIGNN and Matformer exceeds the all-data MAE by more than a factor of five. These failures occur primarily for groups that combine inversion with screws or glides and high cubic or hexagonal symmetry, patterns that are rare among the non-inversion groups used for training. We hypothesize that, in such cases, the GNNs struggle to extrapolate these geometric motifs from related groups, while CFT's explicit symmetry enforcement enables more reliable performance on the unseen groups.

## 6    CONCLUSION

In this work, we introduce the Crystal Fourier Transformer, a single, adaptive neural architecture capable of enforcing invariance to any of the 230 crystallographic space groups. Our approach is built on a novel method for constructing symmetry-adapted Fourier bases by analytically deriving the linear constraints that group operations impose on Fourier coefficients, which are then encoded in a group-conditional routing matrix. We demonstrate that this weight-sharing model learns geometrically meaningful representations of the crystal orbit space, achieves competitive performance on material property prediction benchmarks, and generalizes in a zero-shot setting to unseen space groups, illustrating its potential to overcome the data sparsity inherent to materials science.

ACKNOWLEDGMENTS

The authors would like to thank Keqiang Yan, Kevin Han Huang, Alex Guerra, Drew Novick, Ashwin Sah, and Michael Zhang for helpful discussions and feedback on earlier versions of this work. This work was supported by NSF OAC 2118201 and the Gatsby Charitable Foundation.

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

## A APPENDIX

### A.1 EXAMPLES

There are 17 crystallographic groups in $\mathbb{R}^2$, also called wallpaper groups, which describe the possible symmetries when tiling all of two-dimensional space with a convex shape. The tiling behavior described by each of these groups is displayed in Figure 4.

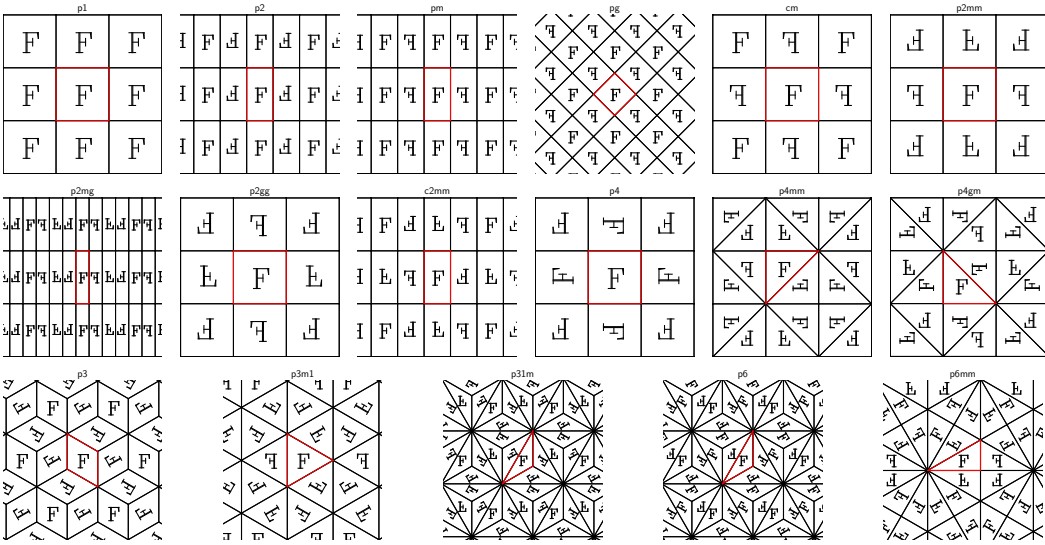

Figure 4: Tiling behavior of the 17 wallpaper groups. The fundamental region $\Pi$, highlighted in red, is the smallest unit that can be repeated to form the entire tiling.

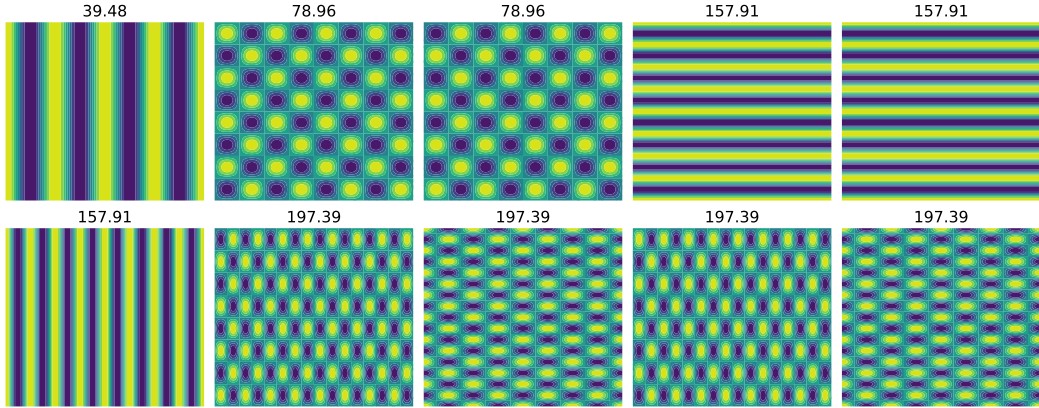

Figure 5: The first 10 non-constant basis functions for the wallpaper group pg. The number displayed above each function is the corresponding eigenvalue from equation 2.

Following the procedure described in Figure 1, we can derive the exact analytic form of the basis functions invariant to any given crystallographic group $G$. By construction, each basis function is of the form

$$e_{\mathcal{O}}(\boldsymbol{x}) = \sum_{\boldsymbol{\omega} \in \mathcal{O}} w_{\boldsymbol{\xi} \to \boldsymbol{\omega}} e^{i 2 \pi \boldsymbol{\omega}^{\top} \boldsymbol{x}},$$

where $\mathcal{O}$ is the orbit of a canonical point $\boldsymbol{\xi}$. This is an eigenfunction of the constrained PDE

$$-\Delta e = \lambda e$$
$$\text{subject to} \quad e = e \circ \phi \text{ for all } \phi \in G \tag{11}$$

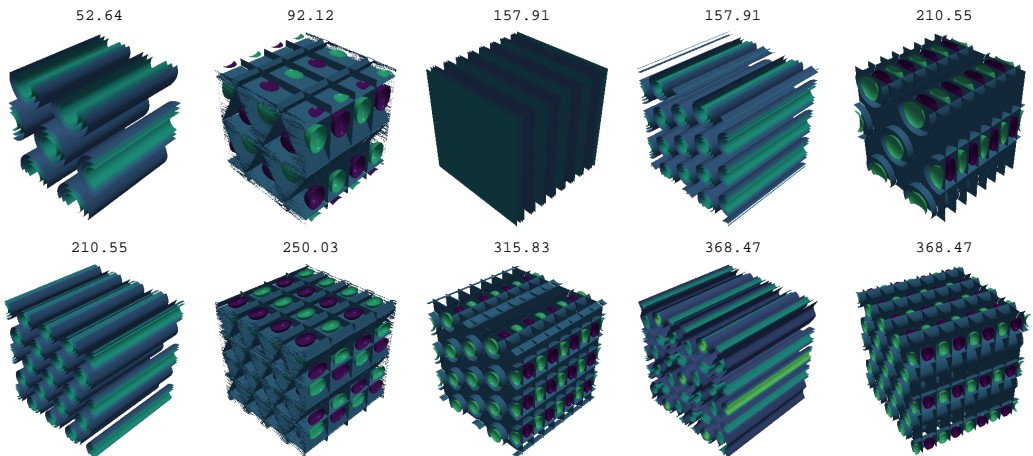

Figure 6: The first 10 non-constant basis functions for space group 193, p6₃/mcm. This group contains 6-fold rotational symmetries, 2-fold rotational symmetries, and mirror symmetries visible along different faces.

with corresponding eigenvalue $4\pi^2||\boldsymbol{\xi}||^2$. The first 10 basis functions for the wallpaper group `pg`, ordered by corresponding eigenvalue, are shown in Figure 5. Our procedure naturally extends to higher-dimensional crystallographic groups; the first 10 basis functions for the space group p6₃/mcm are shown in Figure 6.

## A.2 PROOF OF PROPOSITION 3.1

Consider a function $f : \mathbb{R}^n \to \mathbb{R}$ and its Fourier transform:

$$F(\boldsymbol{\omega}) = \int_{\mathbb{R}^n} f(\boldsymbol{x}) e^{-i2\pi \boldsymbol{\omega}^\top \boldsymbol{x}} d\boldsymbol{x}. \tag{12}$$

Given a crystallographic group $G$ with translations that are integer combinations of $n$ linearly independent basis vectors $\boldsymbol{a}_1, \ldots, \boldsymbol{a}_n$, a $G$-invariant function must be invariant to all translations of the form $\sum_i c_i \boldsymbol{a}_i$ where $\boldsymbol{c} \in \mathbb{Z}^n$, i.e., we require:

$$f\left(\boldsymbol{x} + \sum_i c_i \boldsymbol{a}_i\right) = f(\boldsymbol{x}) \quad \text{for all } \boldsymbol{c} \in \mathbb{Z}^n. \tag{13}$$

Shifting a function introduces a phase factor into the Fourier transform. Satisfying the equation above requires:

$$F(\boldsymbol{\omega}) = e^{i2\pi \boldsymbol{\omega}^\top \sum_i c_i \boldsymbol{a}_i} F(\boldsymbol{\omega}) \quad \text{for all } \boldsymbol{\omega}. \tag{14}$$

For simplicity, let us take $\boldsymbol{a}_1, \ldots, \boldsymbol{a}_n$ to be the standard basis in $\mathbb{R}^n$. Then, in the case where $c_1 = 1$ and all other $c_i = 0$, equation 14 is satisfied if and only if $e^{i2\pi\omega_1} = 1$ or $F(\boldsymbol{\omega}) = 0$. Because $e^{i2\pi\omega_1} = 1$ if and only if $\omega_1 \in \mathbb{Z}$, this simple shift invariance requires that $F(\boldsymbol{\omega}) = 0$ everywhere that $\omega_1 \notin \mathbb{Z}$.

Applying the same logic to each dimension of $\boldsymbol{\omega}$, we see that this shift invariance implies $F(\boldsymbol{\omega}) = 0$ when $\boldsymbol{\omega} \notin \mathbb{Z}^n$, i.e., it can only be nonzero on this discrete grid of points. These points correspond to the Fourier series and the values of $F(\boldsymbol{\omega})$ at those points are the coefficients of the Fourier series. This grid of points is typically referred to as the *reciprocal lattice*.

More generally, our group actions are of the form $\mathbf{A}\boldsymbol{x} + \boldsymbol{t}$, where $\mathbf{A}$ is an orthonormal matrix. We first observe how a change-of-variables of the form $T(\boldsymbol{z}) = \boldsymbol{B}\boldsymbol{z} + \boldsymbol{c}$ alters the Fourier transform.

Taking $\boldsymbol{z} = \boldsymbol{B}^{-1}(\boldsymbol{x} - \boldsymbol{c})$ and $d\boldsymbol{z} = |\boldsymbol{B}|^{-1}d\boldsymbol{x}$:

$$
\begin{aligned}
\int_{\mathbb{R}^n} f(\boldsymbol{B}\boldsymbol{z} + \boldsymbol{c})e^{-i2\pi\boldsymbol{\omega}^\top \boldsymbol{z}}d\boldsymbol{z} &= \frac{1}{|\boldsymbol{B}|}\int_{\mathbb{R}^n} f(\boldsymbol{x})e^{-i2\pi\boldsymbol{\omega}^\top(\boldsymbol{B}^{-1}(\boldsymbol{x}-\boldsymbol{c}))}d\boldsymbol{x} \\
&= \frac{1}{|\boldsymbol{B}|}\int_{\mathbb{R}^n} f(\boldsymbol{x})e^{-i2\pi\boldsymbol{\omega}^\top \boldsymbol{B}^{-1}\boldsymbol{x}}e^{i2\pi\boldsymbol{\omega}^\top \boldsymbol{B}^{-1}\boldsymbol{c}}d\boldsymbol{x} \\
&= \frac{e^{i2\pi\boldsymbol{\omega}^\top \boldsymbol{B}^{-1}\boldsymbol{c}}}{|\boldsymbol{B}|}\int_{\mathbb{R}^n} f(\boldsymbol{x})e^{-i2\pi(\boldsymbol{B}^{-\top}\boldsymbol{\omega})^\top \boldsymbol{x}}d\boldsymbol{x} \\
&= \frac{e^{i2\pi\boldsymbol{\omega}^\top \boldsymbol{B}^{-1}\boldsymbol{c}}}{|\boldsymbol{B}|}F(\boldsymbol{B}^{-\top}\boldsymbol{\omega})
\end{aligned}
$$

Given $f(\mathbf{A}\boldsymbol{x} + \boldsymbol{t}) = f(\boldsymbol{x})$, where $|\mathbf{A}| = 1$ and $\mathbf{A}^{-\top} = \mathbf{A}$, this requires

$$
F(\boldsymbol{\omega}) = e^{i2\pi\boldsymbol{\omega}^\top \mathbf{A}^\top \boldsymbol{t}}F(\mathbf{A}\boldsymbol{\omega}) \quad \text{for all } \boldsymbol{\omega}. \tag{15}
$$

### A.3 PROOF OF THEOREM 3.2

Let $G$ be a crystallographic group and $\mathcal{L}^*$ its reciprocal lattice. By Proposition 3.1, if $f$ is $G$-invariant with Fourier coefficients $F(\boldsymbol{\omega})$, then for any $\phi = (\mathbf{A}, \boldsymbol{t}) \in G$ and $\boldsymbol{\omega} \in \mathcal{L}^*$,

$$
F(\boldsymbol{\omega}) = e^{i2\pi\,\boldsymbol{\omega}^\top \mathbf{A}^\top \boldsymbol{t}}\, F(\mathbf{A}\boldsymbol{\omega}). \tag{16}
$$

This relation implies that the point–group factors $\mathbf{A}$ partition $\mathcal{L}^*$ into disjoint orbits $\mathcal{O} \subset \mathcal{L}^*$.

**Definition A.1** (Phase consistency). *Fix an orbit $\mathcal{O}$ and $\boldsymbol{\omega} \in \mathcal{O}$. The stabilizer of $\boldsymbol{\omega}$ in $G$ is $G_{\boldsymbol{\omega}} := \{(\boldsymbol{A}, \boldsymbol{t}) \in G : \boldsymbol{A}\boldsymbol{\omega} = \boldsymbol{\omega}\}$. We call $\mathcal{O}$* phase-consistent *if*

$$
e^{i2\pi\,\boldsymbol{\omega}^\top \boldsymbol{A}^\top \boldsymbol{t}} = 1 \quad \text{for all } (\boldsymbol{A}, \boldsymbol{t}) \in G_{\boldsymbol{\omega}}.
$$

*Equivalently (via concatenation derived below), every closed path in the constraint graph on $\mathcal{O}$ has edge-weight product 1.*

We encode equation 16 as a directed, edge-weighted graph on $\mathcal{L}^*$: for each $(\boldsymbol{A}, \boldsymbol{t}) \in G$ and $\boldsymbol{\omega} \in \mathcal{L}^*$, place an edge $\boldsymbol{\omega} \to \boldsymbol{A}\boldsymbol{\omega}$ of weight $w(\boldsymbol{\omega} \to \boldsymbol{A}\boldsymbol{\omega}) := e^{i2\pi\,\boldsymbol{\omega}^\top \boldsymbol{A}^\top \boldsymbol{t}}$.

**Lemma A.2** (Path concatenation). *If there are edges $(\boldsymbol{\omega}_1 \to \boldsymbol{\omega}_2)$ of weight $w_1$ and $(\boldsymbol{\omega}_2 \to \boldsymbol{\omega}_3)$ of weight $w_2$, then there is an edge $(\boldsymbol{\omega}_1 \to \boldsymbol{\omega}_3)$ of weight $w_1 w_2$.*

*Proof.* Let $\varphi_1(\boldsymbol{x}) = \boldsymbol{A}_1\boldsymbol{x} + \boldsymbol{t}_1$ and $\varphi_2(\boldsymbol{x}) = \boldsymbol{A}_2\boldsymbol{x} + \boldsymbol{t}_2$. The edge $\boldsymbol{\omega} \to \boldsymbol{A}_1\boldsymbol{\omega}$ has weight $w_1 = e^{i2\pi\boldsymbol{\omega}^\top \boldsymbol{A}_1^\top \boldsymbol{t}_1}$ and $\boldsymbol{A}_1\boldsymbol{\omega} \to \boldsymbol{A}_2\boldsymbol{A}_1\boldsymbol{\omega}$ has weight $w_2 = e^{i2\pi(\boldsymbol{A}_1\boldsymbol{\omega})^\top \boldsymbol{A}_2^\top \boldsymbol{t}_2}$. Since $\varphi_2 \circ \varphi_1(\boldsymbol{x}) = \boldsymbol{A}_2\boldsymbol{A}_1\boldsymbol{x} + (\boldsymbol{A}_2\boldsymbol{t}_1 + \boldsymbol{t}_2)$, the composed edge $\boldsymbol{\omega} \to \boldsymbol{A}_2\boldsymbol{A}_1\boldsymbol{\omega}$ has weight

$$
e^{i2\pi\boldsymbol{\omega}^\top (\boldsymbol{A}_2\boldsymbol{A}_1)^\top(\boldsymbol{A}_2\boldsymbol{t}_1 + \boldsymbol{t}_2)} = e^{i2\pi\left(\boldsymbol{\omega}^\top \boldsymbol{A}_1^\top \boldsymbol{t}_1 + (\boldsymbol{A}_1\boldsymbol{\omega})^\top \boldsymbol{A}_2^\top \boldsymbol{t}_2\right)} = w_1 w_2,
$$

as claimed. □

Fix a phase-consistent orbit $\mathcal{O}$ and a reference $\boldsymbol{\xi} \in \mathcal{O}$. For any $\boldsymbol{\omega} \in \mathcal{O}$, let $p_{\boldsymbol{\xi}\to\boldsymbol{\omega}}$ be a path in the constraint graph from $\boldsymbol{\xi}$ to $\boldsymbol{\omega}$, and define

$$
w_{\boldsymbol{\xi}\to\boldsymbol{\omega}} := \prod_{e \in p_{\boldsymbol{\xi}\to\boldsymbol{\omega}}} w(e).
$$

By Lemma A.2, edge weights multiply under path concatenation. The definition of $w_{\boldsymbol{\xi}\to\boldsymbol{\omega}}$ is independent of the chosen path *iff* every closed cycle has unit product, i.e., it satisfies the phase-consistency condition. If an orbit fails phase-consistency, then traversing a nontrivial cycle gives a non-unit factor, and iterating equation 16 around the cycle forces $F(\boldsymbol{\omega}) = 0$ for all $\boldsymbol{\omega} \in \mathcal{O}$; hence only phase-consistent orbits can support nonzero coefficients.

For a phase-consistent orbit $\mathcal{O}$, choose an arbitrary scalar $c_{\boldsymbol{\xi}} \in \mathbb{C}$ and set

$$
F(\boldsymbol{\omega}) = c_{\boldsymbol{\xi}}\, w_{\boldsymbol{\xi}\to\boldsymbol{\omega}}, \qquad \boldsymbol{\omega} \in \mathcal{O}.
$$

Then equation 16 holds identically on $\mathcal{O}$ (shifting $\boldsymbol{\omega} \mapsto \boldsymbol{A}\boldsymbol{\omega}$ multiplies $w_{\boldsymbol{\xi} \to \boldsymbol{\omega}}$ by the appropriate edge weight), so the inverse Fourier series yields the $G$–invariant function

$$e_{\mathcal{O}}(\boldsymbol{x}) \; = \; \sum_{\boldsymbol{\omega} \in \mathcal{O}} w_{\boldsymbol{\xi} \to \boldsymbol{\omega}} \, e^{i 2\pi \, \boldsymbol{\omega}^\top \boldsymbol{x}}.$$

Distinct orbits $\mathcal{O} \neq \mathcal{O}'$ have disjoint frequency supports, whence $e_{\mathcal{O}} \perp e_{\mathcal{O}'}$ in $L^2(\Pi)$ by Fourier orthogonality on $\Pi$.

Now let $f$ be any $G$–invariant function. Its Fourier coefficients vanish off $\mathcal{L}^*$ and satisfy equation 16 on each orbit. For any orbit $\mathcal{O}$ that is not phase-consistent, the cycle constraint forces $F \equiv 0$ on $\mathcal{O}$; for a phase-consistent $\mathcal{O}$, the coefficients are proportional to $\{w_{\boldsymbol{\xi} \to \boldsymbol{\omega}}\}_{\boldsymbol{\omega} \in \mathcal{O}}$. Therefore

$$f(\boldsymbol{x}) \; = \; \sum_{\mathcal{O} \text{ phase-consistent}} c_{\mathcal{O}} \, e_{\mathcal{O}}(\boldsymbol{x}),$$

i.e., $\{e_{\mathcal{O}}\}$ spans the subspace of $G$–invariant functions in $L^2(\Pi)$. This construction is an explicit realization of the representation theorem of Adams & Orbanz (2023), which guarantees a complete orthogonal basis of constrained Laplace eigenfunctions for continuous $G$–invariant functions.

## A.4 Deriving Fourier Coefficients for Gaussian Density

We show one way in which the symmetry-adapted Fourier basis can encode atom positions in a crystal in a way that captures the atom's environment. Consider a crystal from space group $G$ with a unit cell is defined by $\boldsymbol{B} = [\boldsymbol{b}_1 \quad \boldsymbol{b}_2 \quad \boldsymbol{b}_3]^\top \in \mathbb{R}^{3 \times 3}$ (rows of $\boldsymbol{B}$ are basis vectors). Given an atom in the crystal at position $\boldsymbol{x} \in \mathbb{R}^3$, we want to encode the environment of this atom given by the summation of isotropic Gaussians centered at each atom in the orbit of $\boldsymbol{x}$. Recall that each isometry $\phi \in G$ takes the form $\phi(\boldsymbol{x}) = \mathbf{A}\boldsymbol{x} + \boldsymbol{t}$, where $\mathbf{A}$ is an orthonormal matrix and $\boldsymbol{t}$ is a translation vector. The density of interest is

$$\rho(\mathbf{y}) = \sum_{(\mathbf{A}, \boldsymbol{t}) \in G} \exp\left( -\frac{||\mathbf{y} - (\mathbf{A}\boldsymbol{x} + \boldsymbol{t})||^2}{2\sigma^2} \right). \tag{17}$$

We can approximate this density $\rho$ with a Fourier series. We begin by taking the Fourier transform in terms of $\mathbf{y}$, where

$$\hat{\rho}(\boldsymbol{\omega}) = \int_{\mathbb{R}^3} \rho(\mathbf{y}) e^{-2\pi i \boldsymbol{\omega} \cdot \mathbf{y}} d\mathbf{y}.$$

Recall that for an isotropic Gaussian $g(y; \mu, \sigma) = e^{-||y - \mu||^2 / (2\sigma^2)}$, its Fourier transform is given by

$$\hat{g}(\omega) = (2\pi)^{3/2} \sigma^3 e^{-2\pi^2 \sigma^2 ||\omega||^2 - 2\pi i \omega \cdot \mu}. \tag{18}$$

Our density can be rewritten as

$$\rho(\mathbf{y}) = \sum_{(\mathbf{A}, \boldsymbol{t}) \in G} g(\mathbf{y}; \mathbf{A}\boldsymbol{x} + \boldsymbol{t}, \sigma).$$

By linearity,

$$
\begin{aligned}
\hat{\rho}(\boldsymbol{\omega}) &= \sum_{(\mathbf{A}, \boldsymbol{t}) \in G} (2\pi)^{3/2} \sigma^3 e^{-2\pi^2 \sigma^2 ||\boldsymbol{\omega}||^2 - 2\pi i \boldsymbol{\omega} \cdot (\mathbf{A}\boldsymbol{x} + \boldsymbol{t})} \\
&= (2\pi)^{3/2} \sigma^3 e^{-2\pi^2 \sigma^2 ||\boldsymbol{\omega}||^2} \sum_{(\mathbf{A}, \boldsymbol{t}) \in G} e^{-2\pi i \boldsymbol{\omega} \cdot (\mathbf{A}\boldsymbol{x} + \boldsymbol{t})} \\
&= (2\pi)^{3/2} \sigma^3 e^{-2\pi^2 \sigma^2 ||\boldsymbol{\omega}||^2} \sum_{(\mathbf{A}, \boldsymbol{t}) \in \hat{G}} e^{-2\pi i \boldsymbol{\omega} \cdot (\mathbf{A}\boldsymbol{x} + \boldsymbol{t})} \sum_{\boldsymbol{\ell} \in L} e^{-2\pi i \boldsymbol{\omega} \cdot \boldsymbol{\ell}},
\end{aligned}
$$

where the lattice $L$ is generated by all linear combinations of the basis vectors in $\boldsymbol{b}_1, \boldsymbol{b}_2, \boldsymbol{b}_3$, and $\hat{G}$ is the finite subset of $G$ satisfying

$$\hat{G} = \{ (\mathbf{A}, \boldsymbol{t}) \in G \mid \boldsymbol{t} = t_1 \boldsymbol{b}_1 + t_2 \boldsymbol{b}_2 + t_3 \boldsymbol{b}_3, \; t_i \in [0, 1) \}.$$

Applying Poisson summation to the lattice comb, we have

$$\sum_{\boldsymbol{\ell} \in L} e^{-2\pi i \boldsymbol{\omega} \cdot \boldsymbol{\ell}} = |\boldsymbol{B}| \sum_{\boldsymbol{k} \in L^*} \delta(\boldsymbol{\omega} - \boldsymbol{k}) \tag{19}$$

where $L^*$ denotes the reciprocal lattice. Hence

$$\hat{\rho}(\boldsymbol{\omega}) = (2\pi)^{3/2} \sigma^3 |\boldsymbol{B}| e^{-2\pi^2 \sigma^2 ||\boldsymbol{\omega}||^2} \sum_{\boldsymbol{k} \in L^*} \left[ \sum_{(\mathbf{A}, \boldsymbol{t}) \in \hat{G}} e^{-2\pi i \boldsymbol{\omega} \cdot (\mathbf{A}\boldsymbol{x} + \boldsymbol{t})} \right] \delta(\boldsymbol{\omega} - \boldsymbol{k}). \tag{20}$$

Taking the inverse Fourier transform, we see that the target density $\rho(\mathbf{y})$ can be written as a Fourier series where the coefficient of the term $e^{i2\pi \boldsymbol{k} \cdot \mathbf{y}}$, for $\boldsymbol{k} \in L^*$, is

$$\tilde{\rho}[\boldsymbol{k}] = (2\pi)^{3/2} \sigma^3 e^{-2\pi^2 \sigma^2 ||\boldsymbol{k}||^2} \sum_{(\mathbf{A}, \boldsymbol{t}) \in \hat{G}} e^{-2\pi i \boldsymbol{k} \cdot (\mathbf{A}\boldsymbol{x} + \boldsymbol{t})}. \tag{21}$$

Note that when working converting from fractional to Cartesian coordinates, replace $\boldsymbol{k}_f \in \mathbb{Z}^3$ with $\boldsymbol{k}_c = \boldsymbol{B}^{-\top} \boldsymbol{k}_f$.

## A.5 EXPERIMENT DETAILS

In this section, we describe our experimental setup in more detail. We start with the pretraining setup for the symmetry-adapted positional encodings in Section 5.2, then give configuration details for the material property prediction experiments in Section 5.3 and the zero-shot learning experiments in Section 5.4.

### A.5.1 PRETRAINING POSITIONAL ENCODINGS

The central objective of our pretraining is to map atomic positions into an embedding space where the Euclidean distance between embeddings corresponds to the real-space orbit distance between the atoms. The orbit distance

$$d_G(\boldsymbol{x}_1, \boldsymbol{x}_2) := \min_{\phi_1, \phi_2 \in G} ||\phi_1(\boldsymbol{x}_1) - \phi_2(\boldsymbol{x}_2)||_2. \tag{22}$$

is the minimum Euclidean distance between any two atoms in the orbits of the two initial atoms, considering all symmetry operations of the crystal's space group and periodic boundary conditions.

Our model produces embeddings $\mathbf{e}_1 = f(\mathbf{p}_1, G, \mathbf{L})$ and $\mathbf{e}_2 = f(\mathbf{p}_2, G, \mathbf{L})$. The pretraining loss is the mean squared error (MSE) between the orbit distance and the L2 distance between the corresponding embeddings:

$$\mathcal{L}_{\text{pretrain}} = \left( d_G(\mathbf{p}_1, \mathbf{p}_2) - ||\mathbf{e}_1 - \mathbf{e}_2||_2 \right)^2. \tag{23}$$

The model is designed to explicitly handle crystallographic symmetries and lattice geometry through a two-branch architecture:

- Symmetry-Adapted Position Branch: The model first generates a symmetry-adapted representation from the fractional coordinates. The input fractional positions are converted into a set of Fourier features, $\exp(i \cdot 2\pi \cdot \mathbf{p} \cdot \mathbf{k})$, where $\mathbf{k}$ represents a set of reciprocal lattice vectors. This Fourier representation is then processed by a series of 3 residual blocks to produce a position-based feature vector.
- Lattice Geometry Branch: The $3 \times 3$ lattice vectors are flattened into a 9-dimensional vector and fed into a separate network of 3 residual blocks. This branch learns to capture the scale and geometry of the unit cell, which is essential for converting dimensionless fractional distances into real-space distances.

Each residual block consists of two dense layers, with LayerNorm and ReLU after each layer, that expands the feature vector to length 512 then projects back to length 256. The outputs from the two branches are combined via element-wise multiplication. The resulting vector is then passed through a final MLP to produce the 128-dimensional positional embedding. The model was trained for 200 epochs using the Adam optimizer with a learning rate of 0.0002 and a batch size of 2000.

### A.5.2 MATERIAL PROPERTY PREDICTION

We use 80% training, 10% validation, and 10% test data splits across all property prediction experiments. All models are trained for 500 epochs.

**Crystal Fourier Transformer.** Each atom in a crystal is represented by an input token, which is the sum of a 128-dimensional standard learnable atom embedding based on the atomic number, and a 128-dimensional positional encoding from the pretrained encoding module.

The sequence of input tokens is processed by a series of 3 Transformer blocks. Each block consists of a multi-head self-attention layer and a feed-forward network. These blocks have embedding dimension 128 and 8 attention heads per block. The feed-forward network contains two dense layers, the first one expanding the embedding to dimension 512 before projecting back to dimension 128, using SiLU activation and LayerNorm after each layer. After the final Transformer block, we perform masked mean pooling over the atom tokens to produce a single fixed-size vector representation for the entire crystal. This vector is then passed through a two-layer MLP with hidden dimensions of 2048 and 256 to produce the final scalar prediction.

The model was trained using the AdamW optimizer. The training runs were done with learning rate 0.0001, weight decay 0.0001, and batch size 128.

**Matformer.** We trained Matformer models on the 2025 Materials Project dataset using the code provided by the original authors Yan et al. (2022). We use the AdamW optimizer with a learning rate of 5e-4 and a weight decay of 1e-5, coupled with a OneCycleLR scheduler for learning rate adjustments. The models were trained with a batch size of 64, the maximum given the memory constraints of an NVIDIA L40 GPU. For graph construction, a k-nearest neighbor strategy is used to identify the 12 nearest neighbors within an 8 Å cutoff radius. Each atomic number is mapped to a 92-dimensional embedding using the CGCNN (Xie & Grossman, 2018) atomic features, before a linear transformation maps it to a 128-dimensional feature vector used as input to the first Matformer message-passing layer. The crystal lattice vectors are used in the graph construction, but information about bond angles is not.

**ALIGNN** We also retrain the ALIGNN models using the code provided by the original authors (Choudhary & DeCost, 2021). The model comprises of 4 ALIGNN layers and 4 subsequent GCN layers, both utilizing 256 hidden features. The model processes 92-dimensional CGCNN-style atomic number embeddings for nodes, while bond distances and triplet bond angles were expanded using 80 and 40 radial basis functions, respectively, and embedded into a 64-dimensional feature space. Crystal graphs were also constructed with a k-nearest neighbor strategy to identify the 12 nearest neighbors within an 8 Å cutoff radius. We use the AdamW optimizer, a batch size of 32, a learning rate of 5e-4, and a weight decay of 1e-5, coupled with a OneCycleLR scheduler.

### A.5.3 WHY PRETRAIN?

The standard Fourier encoding seen in Figure 2 is dimensionless, so we need a way to capture the different scales of the crystals in the Materials Project. We pretrain the $G$-invariant encoding module to learn to scale the coefficients of the Fourier basis in a way that captures the geometry and scale of the crystal structures, using the orbit distance setup described in Section 5.2. Table 3 shows that pretraining the positional encodings improves property prediction performance across the board, in comparison to end-to-end training with random initialization of the positional encodings.

When pretraining the positional encoding module, we must choose a cutoff radius for the set of Fourier features $e^{i2\pi\omega^\top x}$ to include, where $\omega$ includes all reciprocal lattice vectors up to a certain radius $R$. In practice we have found that $R = 5$, corresponding to 514 Fourier modes, performs the best on downstream prediction tasks while maintaining a tractable model size. Figure 7 shows the trade-off between the number of Fourier modes used and performance on the orbit distance regression task. This error should go to 0 as we increase the number of basis functions.

| Method | Total Energy eV/atom $\downarrow$ | Band Gap eV $\downarrow$ | Bulk Moduli log GPa $\downarrow$ | Shear Moduli log GPa $\downarrow$ |
|---|---|---|---|---|
| CFT with pretraining | $0.197 \pm 0.009$ | $0.306 \pm 0.006$ | $0.082 \pm 0.008$ | $0.158 \pm 0.011$ |
| CFT without pretraining | $0.207 \pm 0.003$ | $0.338 \pm 0.009$ | $0.134 \pm 0.010$ | $0.215 \pm 0.008$ |

Table 3: Test Mean Absolute Error (MAE) comparisons on the Materials Project dataset. Lower values indicate better performance. Results are reported as mean $\pm$ one standard deviation over 4 runs.

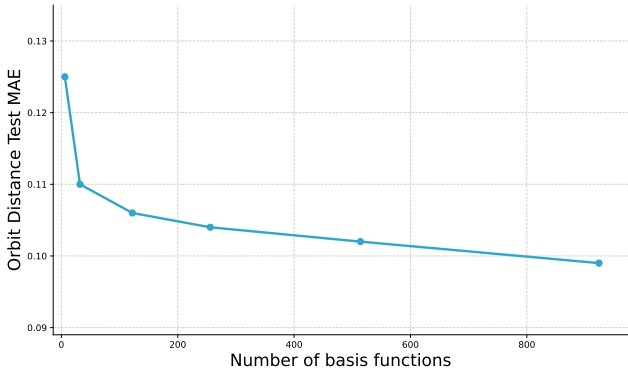

Figure 7: Comparison of the number of Fourier modes used and the resulting test MAE for the orbit distance regression task.

### A.5.4 ZERO-SHOT GENERALIZATION TO UNSEEN GROUPS

In these experiments, we define the hold-out set to be all space groups containing inversion symmetries. This subset represents a significant portion of the data: approximately 49% of the Materials Project dataset for bulk/shear modulus and 25% of the dataset for total energy. The *zero-shot* models are trained exclusively on data from the 138 space groups that do not contain inversion symmetries, whereas the *all-data* models are trained on data from all 230 space groups. Models from both regimes are evaluated on the same dataset of held-out inversion groups to calculate the performance gap.

We use the same CFT architecture as in the previous experiments: 3 Transformer blocks, each consisting of a multi-head self-attention layer and a feed-forward network. These blocks have embedding dimension 128 and 8 attention heads per block. After the final Transformer block, we perform masked mean pooling over the atom tokens and then pass through a two-layer MLP with hidden dimensions of 2048 and 256 to produce the final scalar prediction. Similarly, we use the same Matformer and ALIGNN architectures as we did for the property prediction task, described in Section A.5.2.

## A.6 ORBIT DISTANCES IN P6M

We can visualize the learned embeddings for the 2D wallpaper group p6m. Given a tiling pattern with p6m symmetries and a choice of a canonical fundamental region (e.g., the one highlighted in red on the left in Figure 8), the mapping of each point $x$ to the unique point in its group orbit $\{\phi(x) \mid \phi \in G\}$ that lies in the fundamental region is a known isometric mapping where the standard Euclidean distance perfectly preserves orbit distance. As shown in Figure 8, our model's learned 2D embeddings, using the pretraining setup described in A.5.1, autonomously discover and reproduce this exact geometric mapping without any prior knowledge of it. Different random seeds all produce some rotation of this mapping. This provides strong evidence that our architecture learns the correct underlying metric of the orbit space.

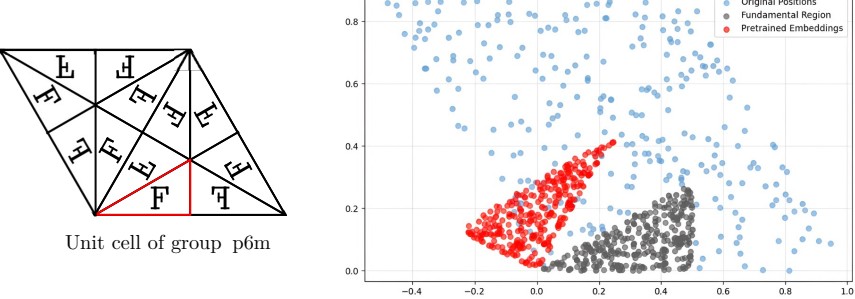

Unit cell of group p6m

Figure 8: In the special case of wallpaper group p6m, which consists of 6-fold rotational symmetry and mirror symmetries, mapping any point to its image in the fundamental region (right triangle bordered in red on the left) exactly preserves the orbit distance between every pair of points. On the right, our $G$-invariant learned positional encodings discover the same isometric mapping in red, without any explicit encoding of this mapping.

