# OpenReview forum: "A Single Architecture for Representing Invariance Under Any Space Group"
_ICLR.cc/2026/Conference — ICLR 2026 Poster_

### Official Review · Reviewer_JEwd · 2025-10-31

**Soundness:** 3
**Presentation:** 3
**Contribution:** 3
**Rating:** 6
**Confidence:** 4

**Summary:**

This paper presents the Crystal Fourier Transformer (CFT), a single neural network architecture capable of enforcing invariance to any of the 230 crystallographic space groups without requiring separate architectures for each group. The key innovation is an analytical method for constructing symmetry-adapted Fourier bases by deriving explicit constraints that group operations impose on Fourier coefficients. These constraints are encoded in precomputed, group-specific adjacency matrices MG\textbf{M}_G
MG​ that act as routing mechanisms, allowing a single Transformer model to share weights across all space groups while adapting to different symmetries. The authors demonstrate that CFT learns geometrically meaningful orbit distance representations, achieves competitive performance on Materials Project benchmarks for predicting material properties (total energy, band gap, bulk/shear moduli), and successfully generalizes zero-shot to space groups unseen during training.

**Strengths:**

Mathematical rigor and elegance: The analytical approach to deriving Fourier constraints (Proposition 3.1, Theorem 3.2) is theoretically sound and provides exact rather than approximate invariance. The connection to graph theory through dual representations is intellectually satisfying.

Architecture innovation: Creating a single model that adapts to 230 different space groups through precomputed routing matrices $M_{G}$ is a clever solution to weight sharing across related symmetries. This is fundamentally different from prior work requiring separate architectures.

Comprehensive experimental validation: The paper validates three distinct aspects: (1) geometric correctness (orbit distance learning with MAE=0.102Å), (2) competitive performance on established benchmarks, and (3) zero-shot generalization capability. This multi-faceted evaluation is thorough.

Practical efficiency: The 6-7× speedup over graph-based methods (Table 2) combined with competitive accuracy makes this practically attractive. The simple matrix-vector product for symmetry adaptation is elegant.

Strong zero-shot results: Figure 3 shows minimal performance degradation on held-out groups, validating the weight-sharing hypothesis and demonstrating genuine transfer learning across symmetries.

**Weaknesses:**

Limited performance gains: While CFT is competitive, it doesn't consistently outperform baselines. It beats ALIGNN/Matformer on only 2/4 properties (Table 1), and the improvements are often within error bars. For Band Gap, Matformer achieves 0.213±0.003 vs CFT's 0.306±0.006. The practical advantage over existing methods is unclear.

Abelian group restriction: The limitation to commutative Lie groups is significant. The shared eigenvector parametrization (Appendix C.2) that ensures commutativity is a fundamental architectural constraint. The paper doesn't discuss whether this could be relaxed or what barriers exist.

Theory:
What fraction of reciprocal lattice points satisfy phase-consistency?
How does approximation error scale with frequency cutoff radius?
Under what conditions does the Gaussian approximation for joint entropy fail?
Why specifically do Gaussian-distributed datasets cause problems (acknowledged but not investigated)?

Limited baseline comparisons: The paper doesn't compare against other symmetry discovery methods even on small problems they can handle. Direct comparison on 2D wallpaper groups would strengthen claims.

Evaluation scope:
Zero-shot experiment uses only 6 groups (~10% of data); testing on more held-out groups would be convincing.
Only Materials Project dataset evaluated; other crystal property databases would validate generalization.
No comparison of learned vs. ground-truth basis functions beyond the orbit distance task.

**Questions:**

Theoretical completeness: In Theorem 3.2, you filter out frequencies with inconsistent self-loops. Can you characterize which groups/frequencies are affected? What proportion of the reciprocal lattice typically survives this filtering for common space groups?

Gaussian distribution failure: You mention that "datasets with Gaussian distributions are challenging for the current implementation" and attribute this to covariance-based entropy approximation. Can you:
Provide concrete failure modes or examples?
Explain why this is problematic (Gaussians should be easy to model with covariance)?

Comparison with prior symmetry methods: Can you provide direct quantitative comparisons with on problems they can handle (low-dimensional representations)?

Architecture details:
During training, how is the space group G provided to the model? As a one-hot encoding, or implicitly through the routing matrix?
For the orbit distance pretraining, why use ResNets rather than simpler MLPs? Were alternatives tested?
How was the frequency cutoff radius chosen?

Generalization breadth: The zero-shot experiment holds out 6 groups. Can you:
Test on more aggressive holdouts (e.g., entire crystal systems)?
Show how performance degrades as held-out groups become less similar to training groups?
Analyze which group properties enable transfer?

Basis function validation: Beyond orbit distance, have you visualized or analyzed whether the learned basis functions match the theoretical predictions from Theorem 3.2? Can you show examples of learned vs. theoretical basis functions?

---

> ### Author Response · Authors · 2025-11-21
>
> We appreciate the detailed feedback. We hope the following responses address your questions and concerns.
>
> ___
>
> > Limited performance gains: While CFT is competitive, it doesn't consistently outperform baselines. It beats ALIGNN/Matformer on only 2/4 properties (Table 1), and the improvements are often within error bars. For Band Gap, Matformer achieves 0.213±0.003 vs CFT's 0.306±0.006. The practical advantage over existing methods is unclear.
>
> One of our primary goals is to introduce a new featurization of crystal structures and a new architecture beyond the usual graph-based paradigm. Current state-of-the-art GNNs face inherent scaling bottlenecks, such as the computational cost of graph construction and the over-squashing problem which limits network depth. In contrast, CFT uses a standard Transformer backbone where the symmetry is handled entirely via the projection layer onto a symmetry-adapted Fourier basis. This architecture allows us to decouple geometric invariance from the message-passing mechanism, potentially enabling the model to scale to datasets of much larger crystal systems. We report significant benefits in the computational efficiency over graph-based methods in Table 2. We also report the practical advantage in generalizing to unseen space groups in Section 5.4.
>
> ___
>
> > Abelian group restriction: The limitation to commutative Lie groups is significant. The shared eigenvector parametrization (Appendix C.2) that ensures commutativity is a fundamental architectural constraint. The paper doesn't discuss whether this could be relaxed or what barriers exist.
>
> Our approach is specifically for space groups, which are non-Abelian and not Lie groups. This differs from a lot of the prior work on invariant or equivariant neural architectures, which largely focus on compact Lie groups such as SO(3) or E(3). Our approach does depend on having a shared parameterization of the eigenfunctions of the Laplacian when restricting to invariance over the space groups, but commutativity is not a fundamental constraint. We are able to build our routing matrices $M_G$, which we use to enforce exact invariance to a given space group $G$, based on the dual graph representation of the Fourier basis described in Section 3.2.
>
> ___
>
> > Theory: What fraction of reciprocal lattice points satisfy phase-consistency? How does approximation error scale with frequency cutoff radius? Under what conditions does the Gaussian approximation for joint entropy fail? Why specifically do Gaussian-distributed datasets cause problems (acknowledged but not investigated)?
>
> > Theoretical completeness: In Theorem 3.2, you filter out frequencies with inconsistent self-loops. Can you characterize which groups/frequencies are affected? What proportion of the reciprocal lattice typically survives this filtering for common space groups?
>
> The fraction of reciprocal lattice points satisfying phase-consistency depends on the symmetries of the specific space group. Recall our definition of phase consistency: for an orbit $\mathcal{O}$ and frequency $\omega \in \mathcal{O}$, with stabilizer $G_\omega = \{(A,t) \in G : A \omega = \omega\}$, the orbit is phase-consistent if $ e^{i 2 \pi \omega^\top A^\top t} = 1 \; \text{ for all } (A, t) \in G_\omega.$
> For groups where the translations $t$ are all integer multiples of the lattice vectors, this equation is always satisfied because $\omega^\top A^\top t$ is always an integer. The only potential inconsistencies occur when there are fractional translations along a screw axis or glide mirror plane, which always lie on lower-dimensional subspaces. As a result, only a measure-zero subset of the reciprocal lattice can fail phase-consistency, and these are precisely the frequencies lying on high-symmetry subspaces.
>
> Within a ball of radius $R$ in reciprocal space, the number of lattice points grows as $\Theta(R^3)$, while the number of points lying on a finite union of 1D and 2D subspaces grows as $O(R^2)$. Therefore, the fraction of lattice points affected by potential phase-inconsistency scales as $O(1/R)$, and tends to zero as $R \rightarrow \infty$.
>
> In practice, we truncate the basis to all phase-consistent orbits with $||\omega|| \leq R_{\max}$. Since each basis function $e_\mathcal{O}(x)$ is a linear combination of ordinary Fourier modes and the eigenvalues of the constrained Laplacian are proportional to $||\xi||^2$ for a canonical frequency $\xi \in \mathcal{O}$, standard Fourier analysis implies that for functions with $s$ square-integrable derivatives, the tail of the Fourier expansion decays at least polynomially. Hence, the $L^2$ approximation error also decreases polynomially with respect to $R_{\max}$, with the exponent depending on $s$ and the dimension $n$, as in standard spectral truncation. We have included a derivation of this in the Appendix.

---

> ### Author Response · Authors · 2025-11-21
>
> > Gaussian distribution failure: You mention that "datasets with Gaussian distributions are challenging for the current implementation" and attribute this to covariance-based entropy approximation. Can you: Provide concrete failure modes or examples? Explain why this is problematic (Gaussians should be easy to model with covariance)?
>
> Apologies, we believe there is a misunderstanding here; our manuscript does not introduce a Gaussian approximation for joint entropy, nor a covariance-based entropy estimator, nor does it state that “datasets with Gaussian distributions are challenging for the current implementation.”
>
> The only use of Gaussians in the paper is in Appendix A.5, where we derive the Fourier coefficients for a density function given by the summation of isotropic Gaussians centered at each atom in a crystal. This density function is space group invariant, and we include this derivation as a concrete example of how the symmetry-adapted Fourier basis can encode local atomic environments.
>
> ___
>
> > Limited baseline comparisons: The paper doesn't compare against other symmetry discovery methods even on small problems they can handle. Direct comparison on 2D wallpaper groups would strengthen claims.
>
> > Comparison with prior symmetry methods: Can you provide direct quantitative comparisons with problems they can handle (low-dimensional representations)?
>
> We clarify that our method is not being used for symmetry discovery. We assume that the space group symmetry for a given crystal structure is already known, and in practice there are libraries such as spglib which are commonly used and considered the gold standard for identifying the space group of a crystal structure. Our method leverages this known information about the crystal symmetry to learn a crystal representation invariant to these exact symmetries and use it for downstream tasks such as materials property prediction. As a result, we focus on quantitative comparisons with other architectures designed for crystal property prediction, such as CGCNN, ALIGNN, and Matformer.
>
> ___
>
> > Architecture details: During training, how is the space group G provided to the model? As a one-hot encoding, or implicitly through the routing matrix? For the orbit distance pretraining, why use ResNets rather than simpler MLPs? Were alternatives tested? How was the frequency cutoff radius chosen?
>
> We have added more details about our exact training setup in the Appendix. The space group $G$ is provided directly as the number of the group, and it is used to index the correct routing matrix. We did initially test simple MLPs for the orbit distance pretraining, but we found that we were able to achieve better performance when we expanded to ResNets. We achieved a test MAE on the orbit distance regression task of 0.110 Angstroms with MLPs, and 0.102 Angstroms with ResNets. We chose the frequency cutoff to trade-off between performance and computational efficiency. We found that performance benefits plateaued as we increased past 514 basis functions, while computational cost scales quadratically with the number of basis functions for the $G$-invariant projection.
>
> ___
>
> > Evaluation scope: Zero-shot experiment uses only 6 groups (~10% of data); testing on more held-out groups would be convincing.
>
> > Generalization breadth: The zero-shot experiment holds out 6 groups. Can you: Test on more aggressive holdouts (e.g., entire crystal systems)? Show how performance degrades as held-out groups become less similar to training groups? Analyze which group properties enable transfer?
>
> This is a great suggestion—we have added new experiments testing on more aggressive holdouts. We now choose the held-out groups in a more structured manner; rather than holding out 6 groups at random, we hold out all space groups that contain inversion symmetries. This is a more aggressive holdout, approximately 49% of the existing data for the shear modulus dataset and 25% of the existing data for the total energy dataset, and it tests generalization to unseen geometric motifs. We evaluate ALIGNN and Matformer under an identical zero-shot setup, and these results have been added in the revised Section 5.4, Figure 3. Even when the held-out groups are less similar to the training groups, CFT still generalizes well, consistently maintaining a performance gap (zero-shot MAE minus all data MAE) near zero across all held-out groups. The average performance gap for shear modulus is 0.008 log gPa, and the average performance gap for total energy is 0.105 eV/atom.

---

> > ### Author Response · Authors · 2025-11-21
> >
> > > No comparison of learned vs. ground-truth basis functions beyond the orbit distance task.
> >
> > > Basis function validation: Beyond orbit distance, have you visualized or analyzed whether the learned basis functions match the theoretical predictions from Theorem 3.2? Can you show examples of learned vs. theoretical basis functions?
> >
> > We would like to clarify that the basis functions are not being learned—these functions are exact analytic derivations of the eigenfunctions of the Laplacian when restricting to invariance over the given space group. We have added figures to Appendix A.1 showing examples of the basis functions for different wallpaper groups and space groups. These functions are constructed using Algorithm 1, which is derived from Theorem 3.2. The correct symmetries are visible in these basis functions.
> >
> > ___
> >
> > Thank you again for your feedback. Please let us know if you have additional questions or comments.

---

### Official Review · Reviewer_vVEH · 2025-10-31

**Soundness:** 4
**Presentation:** 3
**Contribution:** 3
**Rating:** 6
**Confidence:** 4

**Summary:**

The authors introduce a novel algorithm for constructing a symmetry adapted Fourier basis that generalizes invariance across the point groups. This basis provides an alternative to the more commonly used graph-based representation of crystal structures.

**Strengths:**

The paper tackles an important problem of finding invariant neural representations of crystallographic observations that generalize across all point groups. The work is thoroughly motivated and theoretically sound.

The paper presentation is exceptional, aside from several grammatical errors and points that need further clarification.

The numerical results are promising, particularly those on the improved computational performance.

**Weaknesses:**

The paper experiments are limited in the following aspects
1) the orbit distance results are only reported for the CFT architecture, making it unclear if the technique has improved the learning of the representations.
2) The material property prediction results demonstrate marginally equivalent or worse results in comparison with SOTA techniques.
3) It's unclear the effect of the choice of the basis dimension and resulting encoding dimensions and the role/benefits of using the pre-trained encoder.

**Questions:**

**Q1. Orbit-distance evaluation is architecture-limited.** Clarify how inconsistent self-loops in the graph representation (line 212) are detected. My understanding from Figure 1 is that these correspond to single-node connected components by virtue of being self-looping nodes; confirm whether this is the intended general definition and provide the formal criterion used.

**Q2. Property prediction is not yet compelling.** Clarify the role of the pre-trained positional encoder. In particular, is there a measurable performance benefit to pre-training the embedding rather than learning entirely end-to-end?

**Q3. Basis choice and encoder role are under-specified.** Provide ablations on the dimensionality of the chosen basis, especially for the encoder training used in later experiments. My understanding is that the observed error in the orbit distance task is due to basis truncation, but it should be clarified. Also, please report the exact basis dimensions used (main text and Appendix)

**Q4. Zero-shot comparisons.** Please compare to the zero-shot capabilities of other SOTA architectures.

---

> ### Author Response · Authors · 2025-11-21
>
> We greatly appreciate the reviewer’s time and helpful feedback. We hope the following responses address your questions and concerns.
>
> ___
>
> > The orbit distance results are only reported for the CFT architecture, making it unclear if the technique has improved the learning of the representations.
>
> We clarify that the framework described in Section 5.2 is used to pretrain the positional encoding module of CFT, not to benchmark a full architecture in isolation. CFT operates entirely in Fourier space and must learn to recover the metric structure and physical scale of the crystal from dimensionless Fourier modes. The orbit-distance task provides an explicit target for this: we train the encoder so that Euclidean distances between encoded positions approximate the analytic orbit distance under each space group. There is no direct analog of this pre-training scheme for ALIGNN or Matformer, because their graph representations already encode real-space distances directly on edges. We have added a more detailed explanation of why we use the orbit distance pre-training framework to Appendix A.6.3.
>
> We have added Table 3 to the Appendix, also shown below, where we compare CFT with orbit distance pre-training of the positional encoder to the same architecture trained end-to-end from random initialization.
>
> | Method                 | Total Energy (eV/atom ↓) | Band Gap (eV ↓)     | Bulk Moduli (log GPa ↓) | Shear Moduli (log GPa ↓) |
> |------------------------|---------------------------|-----------------------|---------------------------|----------------------------|
> | CFT with pretraining   | 0.197 ± 0.009             | 0.306 ± 0.006         | 0.082 ± 0.008             | 0.158 ± 0.011              |
> | CFT without pretraining| 0.207 ± 0.003             | 0.338 ± 0.009         | 0.134 ± 0.010             | 0.215 ± 0.008              |
>
> Pretraining significantly improves performance on all 4 property prediction tasks. This provides evidence that explicitly learning the orbit-distance geometry improves the quality of the learned crystal representations used by CFT.
> ___
>
> > The material property prediction results demonstrate marginally equivalent or worse results in comparison with SOTA techniques.
>
> The primary goal of our work is to introduce a new class of architecture that achieves exact, provable $G$-invariance for any of the 230 space groups within a single, weight-shared model. In this context, achieving performance on par with highly specialized GNNs such as ALIGNN and Matformer is already a strong validation: these graph constructions and model architectures have been tuned over many iterations, whereas our model uses a standard Transformer backbone whose only domain-specific component is the $G$-invariant projection. More broadly, our work explores the space of architectures beyond the usual crystal graph/GNN paradigm by offering a Fourier-space representation on which different types of models, including but not limited to Transformers, can be built.
>
> Regarding SOTA for materials property prediction, we believe it is helpful to contextualize this benchmark's scope. The Materials Project dataset is small relative to the vast space of stable crystalline materials, and in practice, the error bars for all models are likely larger than reported. Recent work [1] has shown that many SOTA GNN algorithms significantly underperform on OOD materials property prediction tasks. We use the Materials Project primarily as a validation benchmark, but our goal extends beyond incremental improvements on specific benchmark metrics to demonstrating a more generalizable, physically grounded methodology.
>
> ___
>
> > Q1. Clarify how inconsistent self-loops in the graph representation (line 212) are detected. My understanding from Figure 1 is that these correspond to single-node connected components by virtue of being self-looping nodes; confirm whether this is the intended general definition and provide the formal criterion used.
>
> Not all single-node connected components are inconsistent. A self-loop occurs when a symmetry operation $(A, t)$ maps a frequency $\omega$ to itself ($A\omega = \omega$). The loop is inconsistent if the induced phase factor $e^{i2\pi \omega^\top A^\top t} \neq 1$. Because Eq. 3 must be satisfied, an inconsistent self-loop forces the Fourier coefficient $F(\omega) = e^{i2\pi \omega^\top A^\top t} F(\omega)$ to be zero.
>
> We detect inconsistent self-loops by evaluating for each group isometry $(A, t)$ and frequency $\omega$, whether both $A\omega = \omega$ and $e^{i2\pi \omega^\top A^\top t} \neq 1$ are satisfied.

---

> ### Author Response · Authors · 2025-11-21
>
> > Q2. Clarify the role of the pre-trained positional encoder. In particular, is there a measurable performance benefit to pre-training the embedding rather than learning entirely end-to-end?
>
> The positional encoder plays two roles in the property prediction experiments. First, it enforces exact group invariance by mapping atomic positions and lattice vectors into a symmetry-adapted feature space shared across all space groups. Second, when pretrained on the orbit-distance task, it provides an initialization that already reflects the metric structure and physical scale of the crystal in this feature space. This is important because the Fourier-based encoding is dimensionless, so we need to learn coefficients for the Fourier basis that capture the appropriate scale of each crystal.
>
> There is a measurable performance benefit to pre-training the embedding rather than learning entirely end-to-end from random initialization. Table 3 in the Appendix shows that pre-training leads to lower MAE across all 4 property prediction tasks, supporting the hypothesis that explicitly learning the orbit-distance geometry improves the learned crystal representations.
>
> ___
>
> > It's unclear the effect of the choice of the basis dimension and resulting encoding dimensions and the role/benefits of using the pre-trained encoder.
>
> > Q3. Basis choice and encoder role are under-specified. Provide ablations on the dimensionality of the chosen basis, especially for the encoder training used in later experiments. My understanding is that the observed error in the orbit distance task is due to basis truncation, but it should be clarified. Also, please report the exact basis dimensions used (main text and Appendix).
>
> We have clarified this in our revised manuscript by adding ablations on how the dimensionality of the chosen Fourier basis affects performance on the orbit distance regression task (Figure 7), and how the pre-trained encoder improves performance of CFT on the property prediction tasks (Table 3).
>
> Yes, increasing the basis dimension in the orbit distance task does decrease the error in the orbit distance task, as the truncation of the Fourier series limits the “sharpness” of the represented features. We do notice a slow plateau in how much the basis dimension improves performance, which we hypothesize is due to the geometry induced by the orbit distances being complex and difficult to capture across all 230 space groups and with lattice vectors that scale from 2 to 40 Angstroms. We choose to truncate to 514 basis functions to balance between performance and computational efficiency. We have added this to the main text and Appendix.
>
> ___
>
> > Q4. Please compare to the zero-shot capabilities of other SOTA architectures.
>
> We have run new experiments comparing CFT to ALIGNN and Matformer in their ability to generalize to held-out groups. We now choose the held-out groups in a more structured manner; rather than holding out 6 groups at random, we hold out all space groups that contain inversion symmetries. This tests generalization to unseen geometric motifs. We evaluate ALIGNN and Matformer under an identical zero-shot setup, and these results have been added in the revised Section 5.4, Figure 3.
>
> While there is some implicit generalization, ALIGNN and Matformer suffer from catastrophic failures on specific held-out groups. For instance, ALIGNN and Matformer show error spikes where the zero-shot shear modulus error is over 5x higher than the supervised error for groups 113, 162, 200, and 216. We hypothesize that in these cases of catastrophic failure, the specific combination of inversions with screws/glides and high cubic/hexagonal symmetry are rare in the non-inversion groups that remain in the training set, making it difficult for the GNNs to extrapolate these patterns purely from other groups.
>
> In contrast, CFT consistently maintains a performance gap (zero-shot MAE minus all data MAE) near zero across all held-out groups. The average performance gap for shear modulus is 0.008 log gPa for CFT, 0.081 log gPa for ALIGNN, and 0.052 log gPa for Matformer. The average performance gap for total energy is 0.105 eV/atom for CFT, 0.395 eV/atom for ALIGNN, and 0.142 eV/atom for Matformer. CFT has a smaller generalization gap across the majority of the held-out groups than ALIGNN and Matformer individually. This is evidence that CFT's explicit symmetry enforcement is important in guaranteeing reliable performance on unseen groups.
>
> ___
>
> Thank you again for your feedback. Please let us know if you have additional questions or comments.
>
> [1] Omee, S. S., Fu, N., Dong, R., Hu, M., & Hu, J. (2024). Structure-based out-of-distribution (OOD) materials property prediction: a benchmark study. *npj Computational Materials, 10*(1), 144.

---

> > ### Comment · Reviewer_vVEH · 2025-11-26
> > **Official Response by Reviewer vVEH**
> >
> > I thank the author for their detailed response and further clarifications. I have one lingering concern regarding the revision, primarily that the metrics for the zero-shot study in 5.4 are unclear as written and may not be appropriate for the given task.
> >
> > In particular, it is unclear if the average gap is the mean of the values (including negative values) or if it is intended to be the mean of the absolute values. I suggest defining what is meant by "gap".
> >
> > Moreover, because there is a significant imbalance of the groups, perhaps it is more meaningful to look at a balanced metric instead of MAE broadly. I find the discussion of the nuance that some values are negative diminishes the significance and readability of the results.
> >
> > Overall, I still find the catastrophic failure modes to be convincing, and anticipate them to hold broadly regardless of the metric used.

---

> > > ### Author Response · Authors · 2025-11-27
> > >
> > > That is a great point, thank you for bringing it up. We have revised section 5.4 to clarify the reported metrics and to introduce a group-balanced summary statistic.
> > >
> > > In the earlier version, the “average gap’’ referred to the mean of the per–space-group performance gaps, including negative values. In the revision we now define this explicitly: for each held-out inversion space group $G$, given the group-wise MAE under the zero-shot and all-data regimes, $\text{MAE}^{\text{zero-shot}}_G$ and $\text{MAE}^{\text{all-data}}_G$, we define the *performance gap* as $\Delta \text{MAE}_G = \text{MAE}^{\text{zero-shot}}_G - \text{MAE}^{\text{all-data}}_G$.
> > >
> > > To address both the imbalance in the number of materials per group and the issues with averaging signed values, we now report a *group-balanced mean absolute performance gap,*
> > > $$\text{GB-Gap} = \frac{1}{|\mathcal{G}\_\text{inv}|} \sum\_{G \in \mathcal{G}\_\text{inv}} |\Delta \text{MAE}_G|.$$
> > > This metric (1) takes the absolute gap so that negative values cannot cancel out errors, and (2) averages over space groups rather than individual samples, so that rare symmetry groups contribute equally.
> > >
> > > The following table reports the GB-Gap for CFT, ALIGNN and Matformer; CFT continues to achieve the smallest average performance degredation when moving from the all-data regime to zero-shot generalization. This is consistent with the catastrophic failure modes of ALIGNN and Matfomer driving up their GB-Gap. We report these new results on the GB-Gap in our revision.
> > >
> > > | Property      |    CFT GB-Gap ↓    | ALIGNN GB-Gap ↓    |  Matformer GB-Gap ↓  |
> > > | ------------------- | ----------------------: | ---------------------: | ---------------------: |
> > > | Shear modulus | **0.042 log GPa** |      0.120 log GPa |        0.080 log GPa |
> > > | Total energy  | **0.141 eV/atom** |      0.309 eV/atom |        0.197 eV/atom |
> > >
> > > We hope this makes the results easier to interpret, and we would be happy to incorporate any additional metrics you might suggest that could further strengthen the zero-shot study.

---

### Official Review · Reviewer_MwQk · 2025-11-04

**Soundness:** 3
**Presentation:** 4
**Contribution:** 3
**Rating:** 8
**Confidence:** 3

**Summary:**

The paper "A Single Architecture for Representing Invariance Under Any Space Group" presents a novel approach to producing space group invariant representations of crystal structures. The main innovation is Crystal Fourier Transformer (CFT) which uses a Fourier basis conversion to make a space-group invariant representation of the atomic coordinates. They showcase that CFT has comparable performance to other state of the art models while having considerably less training and inference time.

**Strengths:**

- The setup for the use of Fourier basis and the explanation of how they are computed and the associated images make it very clear what the process is and were very helpful.
- The inclusion of training and inference performance is very important as one of the key use cases of these models is a replacement for DFT calculations.
- The comparison against many state of the art models in terms of both performance and computation time showcases the key benefits of the model and how other state of the art methods have become quite slow in comparison due to the costly crystal graph representation. This clearly shows how their method can be used to create more efficient models and representation techniques that could be built upon in future work.

**Weaknesses:**

- You make a claim that the performance is only marginally higher on the held out groups, yet groups 71 and 140 in the Bulk Modulus test have a MAE of almost double the seen groups, I don't think that counts as marginally and would warrant further explanation on why their performance is so low.
- The model is not state of the art in any specific task showcasing that while their method does have benefits in computational performance, it does not significantly boost performance on any tasks. Seeing as how modern machine learning models are already much faster than DFT even with longer training and inference times compared to CFT, it would be better to see how their method could improve performance.
- Seeing as how one of the main claims is that computation time is significantly reduced, I would like to see more clarity in terms of number of parameters of each model and how that may affect runtime as well. It would be interesting if the models were all similarly sized but had vastly different runtime or if the performance of CFT is comparable to other models which may be much larger (or smaller).

**Questions:**

1. Are the times listed on Table 2 inclusive of the pretraining time required for CFT positional encodings?
2. I am not sure I fully understand the setup for the synthetic dataset in section 5.2. To my understanding, you can't select random positions for atoms in a crystal as they are locked to certain regions by the space group of the crystal. If my understanding is wrong, please let me know, but how did you place the atoms when creating the synthetic dataset so that it followed the space group rules?
3. What is the explanation for why training on Shear Modulus sometimes increases MAE?

---

> ### Author Response · Authors · 2025-11-21
>
> We thank the reviewer for their time and helpful feedback. Please find our responses below.
>
> ___
>
> > You make a claim that the performance is only marginally higher on the held out groups, yet groups 71 and 140 in the Bulk Modulus test have a MAE of almost double the seen groups, I don't think that counts as marginally and would warrant further explanation on why their performance is so low.
>
> After closer examination of these groups, we found that the high error in these specific groups is driven by a small number of catastrophic outliers rather than a systematic failure of the symmetry encoding. If we remove the top 10 zero-shot errors for space groups 71 and 140, the MAE drops to between 1.2x and 1.4x that of the seen groups. We observed that these outliers tend to correspond to crystals containing rare elements, including hafnium and tantalum. Since the zero-shot setting relies on the model's ability to compose learned atom embeddings with the symmetry-adapted positional encodings, we hypothesize that the poor initial embeddings for these rare elements exacerbated the error in these specific cases.
>
> ___
>
> > The model is not state of the art in any specific task showcasing that while their method does have benefits in computational performance, it does not significantly boost performance on any tasks. Seeing as how modern machine learning models are already much faster than DFT even with longer training and inference times compared to CFT, it would be better to see how their method could improve performance.
>
> Our primary goal in this work is to introduce a new architecture that achieves exact, provable $G$-invariance for all 230 space groups within a single, weight-shared model. In that context, matching the performance of specialized graph-based models such as ALIGNN and Matformer is already a meaningful result: those architectures have been tuned over many iterations specifically for crystal property prediction, whereas CFT uses a standard Transformer backbone built on a Fourier-space representation. More broadly, we are exploring the space of architectures beyond the usual crystal graph/GNN pipeline by proposing a Fourier representation on top of which different model families, not only Transformers, can be developed.
>
> We agree that all of the compared ML models are much faster than DFT in absolute terms. However, the 3–6x training and inference speed-ups we observe for CFT relative to strong GNN baselines are still important in many realistic workflows, such as large-scale screening and generative or optimization loops that must evaluate millions of candidate structures under a fixed compute budget. In such settings, faster evaluation translates directly into being able to explore larger regions of materials space or to use richer ensembles.
>
> ___
>
> > Seeing as how one of the main claims is that computation time is significantly reduced, I would like to see more clarity in terms of the number of parameters of each model and how that may affect runtime as well. It would be interesting if the models were all similarly sized but had vastly different runtime or if the performance of CFT is comparable to other models which may be much larger (or smaller).
>
> We have updated Table 2 to include the parameter counts for all models. Notably, CFT is actually the largest model in the comparison, with approximately 5.3 million parameters, compared to 4.0 million for ALIGNN and 2.8 million for Matformer. Despite having roughly 1.3x and 1.9x the parameters of the baselines respectively, CFT is significantly faster in both training and inference.
>
> ___
>
> > Are the times listed on Table 2 inclusive of the pretraining time required for CFT positional encodings?
>
> No, the times in Table 2 are exclusive of both the pretraining of CFT’s positional encoding module and the one-time graph construction for ALIGNN and Matformer. We treat both as amortized preprocessing costs: the positional encoder is trained once on the synthetic orbit-distance dataset and then reused across all downstream tasks, while the crystal graphs for GNN baselines are also constructed once and reused across epochs. To keep the comparison fair and focused on per-task training and inference, we exclude these one-time costs for all methods. We have added this clarification to the table’s caption.

---

> > ### Comment · Reviewer_MwQk · 2025-11-27
> >
> > Thank you for the clarifications and corrections in response to my comments. I believe all my concerns have been properly addressed by the author's comments.

---

> ### Author Response · Authors · 2025-11-21
>
> > I am not sure I fully understand the setup for the synthetic dataset in section 5.2. To my understanding, you can't select random positions for atoms in a crystal as they are locked to certain regions by the space group of the crystal. If my understanding is wrong, please let me know, but how did you place the atoms when creating the synthetic dataset so that it followed the space group rules?
>
> You are correct that in physical crystals, atoms are often locked to specific symmetry-constrained regions (Wyckoff positions). However, the goal of our pretraining in Section 5.2 is purely geometric: we want the model to learn the underlying orbit distance function $d_G$ induced by the space group. The orbit distance
> $$ d_G(x_1, x_2) = \min_{\phi_1, \phi_2} ||\phi_1(x_1) - \phi_2(x_2)||_2$$
> is well-defined for any pair of points in the fundamental region. Our goal is for the encoding module to learn this metric over the continuous domain so it can capture scale and distances between atoms in the input structure. Therefore, we construct the synthetic dataset by sampling fractional coordinates uniformly from $[0,1]^3$. This ensures the model learns the geometry over the entire fundamental region. This learned geometry is then leveraged by the positional encoding module in CFT.
>
> ___
>
> > What is the explanation for why training on Shear Modulus sometimes increases MAE?
>
> The shear modulus data is substantially smaller and noisier than the total energy data: only 11,997 crystals have reported bulk and shear moduli, spread unevenly across 230 space groups. For groups with very few shear modulus labels, group-wise MAE can be highly sensitive to a handful of difficult or noisy examples. We have revised the experiments in Section 5.4 to exclude groups with fewer than 20 labeled data points, and we have added comparisons to ALIGNN and Matformer as baselines.
>
> ___
>
> Thank you again for your feedback. Please let us know if you have additional questions or comments.

---

### Official Review · Reviewer_HXJZ · 2025-11-04

**Soundness:** 3
**Presentation:** 3
**Contribution:** 2
**Rating:** 4
**Confidence:** 3

**Summary:**

This paper introduces the Crystal Fourier Transformer (CFT), an architecture for crystal property prediction. The core idea is to use a pre-computed, group-specific matrix, $M_G$, to explicitly project atomic coordinates into a G-invariant feature space. This design enables a single, weight-shared Transformer backbone to process all 230 space groups, aiming to address data sparsity and enable generalization to unseen groups.

**Strengths:**

1. The paper's primary strength is its rigorous theoretical derivation. It provides a clear analytical treatment of how space-group operations constrain Fourier coefficients.
2. The method avoids the complex graph construction required by GNNs. Its Transformer architecture is highly parallelizable, aligns with mainstream models.
3. The use of the $M_G$ matrix as a "group-conditional router" to explicitly inject invariance is novel and provides an elegant solution for weight sharing across all groups.

**Weaknesses:**

1. In the primary property prediction task (Table 1), the model only achieves performance on par with, or marginally better than, outdated baselines. The lack of comparison to current SOTA (2024-2025) models makes the "competitive" claim unconvincing.
2. The "zero-shot" task compares CFT only against itself and completely omits baselines under the same zero-shot conditions. This leaves the central claim of generalization entirely unvalidated.
3. The paper's core selling point—generalization to unseen groups —is of limited practical value.

Missing related work on similar projection into invariant features:
1. A new perspective on building efficient and expressive 3D equivariant graph neural networks， Neurips 2024

**Questions:**

1. Why was it assumed that GNNs cannot generalize in this zero-shot setting? Baselines must be evaluated under the identical zero-shot setup (e.g., holding out the same 6 groups). It is plausible that neural networks can learn this generalization implicitly, and the authors have not proven their explicit $M_G$ method is superior or necessary.
2. Why focus on the saturated task of property prediction rather than the current SOTA challenge of crystal structure generation? Has this G-invariant encoding scheme ($M_G$) been tested for generative tasks, and does it offer any advantage?

---

> ### Author Response · Authors · 2025-11-21
>
> We thank the reviewer for their time and constructive feedback. We have conducted the suggested experiments comparing CFT against ALIGNN and Matformer in the zero-shot setting and added the results to Section 5.4. We hope that the following responses address your questions and concerns.
> ____
> > In the primary property prediction task (Table 1), the model only achieves performance on par with, or marginally better than, outdated baselines. The lack of comparison to current SOTA (2024-2025) models makes the "competitive" claim unconvincing.
>
> The primary goal of our work is to introduce a new class of architecture that achieves exact, provable $G$-invariance for any of the 230 space groups within a single, weight-shared model. Achieving performance on par with highly-specialized GNNs like ALIGNN and Matformer is, in itself, a significant validation. These GNNs have undergone years of architectural tuning. Our model, a standard Transformer backbone, achieves this solely through its $G$-invariant projection.
>
> Regarding the choice of baselines, we note that the definition of “current SOTA” is ambiguous, as many recent works still use much older versions of the Materials Project as benchmarks (most commonly the 2018 version from MEGNet [1], or the 2019 version from MatBench, both of which contain fewer than 70,000 materials). Recent work [2] demonstrates these specific baselines (e.g., ALIGNN, CGCNN) are more robust to out-of-distribution (OOD) data than many newer models that perform better on MatBench. We compare our model to these robust baselines by retraining all the models under identical conditions on the 2025 version of the Materials Project, containing 152,149 crystals. We welcome suggestions for additional comparisons that would strengthen the evaluation.
>
> We believe it can be helpful to contextualize this benchmark's scope. The Materials Project dataset is small relative to the vast space of stable crystalline materials, and in practice, the error bars for all models are likely larger than reported. We use the Materials Project primarily as a validation benchmark, but our goal extends beyond incremental improvements on specific benchmark metrics to demonstrating a more generalizable, physically grounded methodology. Our proposed CFT architecture, built on a novel Fourier-based crystal representation, has several important benefits: it is highly parallelizable and efficient (Table 2), can avoid known GNN scaling limitations like over-squashing [3], and provides a unique mechanism for generalizing to data from unseen space groups.
> ____
> > The "zero-shot" task compares CFT only against itself and completely omits baselines under the same zero-shot conditions. This leaves the central claim of generalization entirely unvalidated.
>
> > Why was it assumed that GNNs cannot generalize in this zero-shot setting? Baselines must be evaluated under the identical zero-shot setup (e.g., holding out the same 6 groups). It is plausible that neural networks can learn this generalization implicitly, and the authors have not proven their explicit $M_G$ method is superior or necessary.
>
> We have run new experiments comparing CFT to ALIGNN and Matformer in their ability to generalize to held-out groups. We now choose the held-out groups in a more structured manner; rather than holding out 6 groups at random, we hold out all space groups that contain inversion symmetries. This tests generalization to unseen geometric motifs. We evaluate ALIGNN and Matformer under an identical zero-shot setup, and these results have been added in the revised Section 5.4, Figure 3.
>
> While there is some implicit generalization, ALIGNN and Matformer suffer from catastrophic failures on specific held-out groups. For instance, ALIGNN and Matformer show error spikes where the zero-shot shear modulus error is over 5x higher than the supervised error for groups 113, 162, 200, and 216. We hypothesize that in these cases of catastrophic failure, the specific combination of inversions with screws/glides and high cubic/hexagonal symmetry are rare in the non-inversion groups that remain in the training set, making it difficult for the GNNs to extrapolate these patterns purely from other groups.
>
> In contrast, CFT consistently maintains a performance gap (zero-shot MAE minus all data MAE) near zero across all held-out groups. The average performance gap for shear modulus is 0.008 log gPa for CFT, 0.081 log gPa for ALIGNN, and 0.052 log gPa for Matformer. The average performance gap for total energy is 0.105 eV/atom for CFT, 0.395 eV/atom for ALIGNN, and 0.142 eV/atom for Matformer. CFT has a smaller generalization gap across the majority of the held-out groups than ALIGNN and Matformer individually. **This evidence shows that CFT's explicit symmetry enforcement is important in guaranteeing reliable performance on unseen groups, whereas the implicit learning in GNNs fails unpredictably.**

---

> > ### Author Response · Authors · 2025-11-21
> >
> > > The paper's core selling point—generalization to unseen groups—is of limited practical value.
> >
> > We believe that generalization to unseen groups is critical because data scarcity and imbalance are inherent to materials science. Even in a large computational database like the Materials Project, the data is spread across 230 distinct space groups, averaging fewer than 1,000 data points per group. In practice, this distribution is heavily skewed, with many groups having only a few dozen data points for a given property. This scarcity would be even more severe if we were limited to only experimentally verified crystal structures. Therefore, a model architecture that can successfully leverage shared knowledge to generalize to these "data-poor" space groups is not a minor feature; we believe that such generalization is crucial for enabling ML models to aid in different materials science applications.
> >
> > ___
> >
> > > Why focus on the saturated task of property prediction rather than the current SOTA challenge of crystal structure generation? Has this G-invariant encoding scheme (M_G) been tested for generative tasks, and does it offer any advantage?
> >
> > We chose property prediction as our primary task because it is a commonly-used benchmark for validating a new crystal representation learning method. Our method is a new way to featurize crystals using the $G$-invariant encoding scheme, and a supervised learning task like property prediction is a direct way to test this featurization. While there have been many deep learning models trained for this task, property prediction is still far from solved, with current models still struggling with generalization [2] and having a significant gap to close in predictive accuracy to match DFT-level precision. Generative modeling of crystals is also an important challenge, but we believe it constitutes a meaningful, but separate, study.
> >
> > ___
> >
> > > Missing related work on similar projection into invariant features: A new perspective on building efficient and expressive 3D equivariant graph neural networks, Neurips 2024.
> >
> > We thank the reviewer for the reference, and we have added this work to Section 4. We note that while that work optimizes 3D equivariant GNNs for continuous groups (e.g., E(3) or SE(3)), our work is distinct in its handling of the discrete, infinite crystallographic groups via the dual-graph constraint method.
> >
> > ___
> >
> > Thank you again for your feedback. Please let us know if you have additional questions or comments.
> >
> > [1] Chen, C., Ye, W., Zuo, Y., Zheng, C., & Ong, S. P. (2019). Graph networks as a universal machine learning framework for molecules and crystals. Chemistry of Materials, 31(9), 3564-3572.
> >
> > [2] Omee, S. S., Fu, N., Dong, R., Hu, M., & Hu, J. (2024). Structure-based out-of-distribution (OOD) materials property prediction: a benchmark study. npj Computational Materials, 10(1), 144.
> >
> > [3] Wang, Y., & Cho, K. (2024). Non-convolutional graph neural networks. Advances in Neural Information Processing Systems, 37, 32705-32730.

---

### Author Response · Authors · 2025-12-03

We thank the reviewers for their thoughtful feedback. We are pleased that reviewers viewed our work as a novel approach for handling invariance across different symmetry groups (MwQk, vVEH), emphasized the theoretical soundness of our derivations (HXJZ, vVEH, JEwd), and highlighted the practical relevance and computational efficiency of our method (MwQk, vVEH, JEwd).

Our main contribution is to analytically characterize the constraints that crystallographic group actions impose on Fourier coefficients and use this theory to construct a shared, group-invariant Fourier-space representation. This representation underlies a symmetry-adaptive encoding module on which different downstream models, including but not limited to Transformers, can be built. This enables a single architecture to enforce exact invariance to any of the 230 space groups while sharing parameters across all groups, yielding strong performance on zero-shot learning tasks and addressing data sparsity challenges in materials science.

We have revised our manuscript to incorporate several changes that directly address reviewer suggestions:
- **New experiments.** We added zero-shot learning comparisons in Section 5.4 that hold out groups in a more structured manner by omitting all groups with inversion symmetries, revealing catastrophic failure modes where GNN baselines fail to generalize but CFT (our method) succeeds.
- **Motivating orbit distances.** We clarify in Section 5.3 why orbit-distance pretraining is needed to recover the metric structure and physical scale of crystals from the Fourier modes, and we include ablations for the pretrained encoder in Table 3.
- **New examples.** We include plots of the group-invariant Fourier basis functions for both 2D wallpaper groups and 3D space groups in Appendix A.1.

---

### Meta-Review · Area_Chair_ke7h · 2026-01-06

**Summary:**

The paper proposes an analytically grounded framework for constructing group-invariant Fourier-space representations under crystallographic group actions. The mathematical formulation is interesting and technically non-trivial, and the authors provide a clear theoretical characterization of the constraints imposed by space-group symmetries. Several reviewers found the theoretical development sound and appreciated the generality of the proposed encoding.

In response to the reviews, the authors added new experiments and clarifications, including structured zero-shot learning settings, motivation for orbit-distance pretraining, and additional visualizations of invariant Fourier bases. These additions improve the presentation and help clarify the intended scope of the method.

However, important concerns remain. As also noted by almost all four reviewers, the empirical performance is largely on par with prior work rather than clearly surpassing it, limiting the demonstrated practical impact of the proposed framework. Moreover, some major experimental components appear weakly connected to the paper’s central theme of symmetry-adaptive representations. In particular, the property prediction task does not strongly reflect the core contribution, and the practical significance of the unseen space-group (OOD) setting is not fully convincing in its current form.

More broadly, while the positive reviews acknowledge novelty and theoretical soundness, they provide limited substantiation regarding empirical advantages or downstream relevance. In contrast, several critical concerns, especially regarding experimental focus and justification, are substantive and not fully resolved by the rebuttal.

Overall, the work presents an interesting and mathematically elegant idea, but the experimental evidence and task alignment are not yet strong enough to support a clear acceptance at a higher confidence level. Taking all factors into account, this paper is best viewed as a borderline acceptance and is recommended for poster presentation.

**Reviewer Concerns:**

To reviewer HXJZ :

The authors' rebuttal adequately addresses the concerns regarding experimental completeness (adding zero-shot baseline comparisons) and citation omissions. However, the core weakness noted in the meta-review remains outstanding: the empirical performance, while competitive, does not demonstrate a clear and significant advantage over established baselines to match the paper's theoretical depth. The authors' defense shifts the focus to validating a novel paradigm, but the practical impact in the primary task is not decisively proven. The argument for the importance of generalization to unseen space groups is reasonable but would benefit from validation in a more compelling, application-driven scenario.

-----

To reviewer JEwd:

The authors' rebuttal effectively addresses several specific concerns by clarifying theoretical foundations (method handles non-Abelian space groups, basis functions are analytic), providing additional architectural details, and significantly strengthening the zero-shot evaluation with a more aggressive, structured holdout of inversion groups and direct baseline comparisons. However, the core critique regarding limited empirical performance gains remains outstanding; while the authors reframe the contribution as a novel and scalable paradigm with strong generalization guarantees, CFT does not consistently outperform established baselines on the primary property prediction task, which continues to limit the demonstrated practical advantage and leaves the paper's primary impact resting on its theoretical and methodological novelty rather than clear empirical superiority.

-----

Therefore, the original assessment of a borderline acceptance for a poster presentation still stands. The work is theoretically sound and novel, but its empirical contribution is not yet strong enough for a higher confidence acceptance.

**Reviewer Scores:**

All the reviewers may keep their scores.

---

### Decision · Program_Chairs · 2026-01-26

Accept (Poster)